# The folate cycle enzyme MTHFD2 induces cancer immune evasion through PD-L1 up-regulation

Man Shang[1,8], Huijie Yang[1,8], Ran Yang[1], Tao Chen[2], Yuan Fu[1], Yeyi Li[1], Xianlong Fang[3], Kangjian Zhang[3,4], Jianju Zhang[5], Hui Li[3], Xueping Cao[3], Jinfa Gu[3,4], Jianwen Xiao[3], Qi Zhang[6], Xinyuan Liu[4], Qiujing Yu [7✉] & Ting Wang [1✉]

Metabolic enzymes and metabolites display non-metabolic functions in immune cell signalling that modulate immune attack ability. However, whether and how a tumour's metabolic remodelling contributes to its immune resistance remain to be clarified. Here we perform a functional screen of metabolic genes that rescue tumour cells from effector T cell cytotoxicity, and identify the embryo- and tumour-specific folate cycle enzyme methylenetetrahydrofolate dehydrogenase 2 (MTHFD2). Mechanistically, MTHFD2 promotes basal and IFN-γ-stimulated PD-L1 expression, which is necessary for tumourigenesis in vivo. Moreover, IFN-γ stimulates MTHFD2 through the AKT–mTORC1 pathway. Meanwhile, MTHFD2 drives the folate cycle to sustain sufficient uridine-related metabolites including UDP-GlcNAc, which promotes the global O-GlcNAcylation of proteins including cMYC, resulting in increased cMYC stability and PD-L1 transcription. Consistently, the O-GlcNAcylation level positively correlates with MTHFD2 and PD-L1 in pancreatic cancer patients. These findings uncover a non-metabolic role for MTHFD2 in cell signalling and cancer biology.

---

[1] Department of Pharmacology, Tianjin Key Laboratory of Inflammatory Biology, The province and ministry co-sponsored collaborative innovation center for medical epigenetics, School of Basic Medical Sciences, Tianjin Medical University, Tianjin, China. [2] Endoscopy Center, Shanghai East Hospital, Tongji University, Shanghai, China. [3] Shanghai Yuansong Bio-technology Limited Company, Shanghai, China. [4] State Key Laboratory of Cell Biology, Shanghai Institute of Biochemistry and Cell Biology, Center for Excellence in Molecular Cell Science, Chinese Academy of Sciences, Beijing, China. [5] Department of Pediatric Surgery, Tianjin Medical University General Hospital, Tianjin, China. [6] Tianjin Key Laboratory of Acute Abdomen Disease Associated Organ Injury and ITCWM Repair, Institute of Integrative Medicine for Acute Abdominal Diseases, Integrated Chinese and Western Medicine Hospital (Nankai Hospital), Tianjin University, Tianjin, China. [7] Key Laboratory of Immune Microenvironment and Disease (Ministry of Education), Department of Immunology, School of Basic Medical Sciences, Tianjin Medical University, Tianjin, China. [8] These authors contributed equally: Man Shang, Huijie Yang. ✉email: yuqiujing2018@tmu.edu.cn; twang1@tmu.edu.cn

Metabolic reprogramming is considered a hallmark of both tumor and immune development. There is evidence that metabolites function as regulators for signaling proteins, either through direct interaction or by modulating their post-translational modifications as substrate. Therefore, in addition to bioenergetic and biosynthetic supply, the rewired metabolic pathways also possess roles in signaling, which influence various cellular activities. For example, succinate, fumarate, itaconate, 2-hydroxyglutarate isomers, acetyl-CoA, lactate, and kynurenine, which are mainly produced by immune cell itself and some are released from tumor cell like lactate, have been reported to target immune cell signaling, which might affect tumor-killing[1,2]. On the other side, whether and how the tumor rewired metabolic pathways, which support their growth and proliferation, could also directly stimulate the signals in tumor cells that empower them to resist immune attack remain to be explored.

The folate cycle, also called one-carbon metabolism, mainly supports cellular nucleotide supply, amino acid (glycine and serine) homeostasis, and S-adenosylmethionine (SAM) production[3]. The folate reactions are subcellularly compartmentalized. In mitochondria, two separate reactions are catalyzed by bifunctional dehydrogenase/cyclohydrolase enzyme (MTHFD2/2L) and formate-THF ligase enzyme (MTHFD1L), whereas the same reactions are performed by MTHFD1, a single trifunctional enzyme in the cytosol[4].

The folate cycle is efficiently driven by MTHFD2 in embryonic tissue to meet the high nucleotide demand required to support cell proliferation. In contrast, in adult tissues, the cycle is maintained at a basal level only by MTHFD2L, the $K_{cat}/K_M$ for which is 50-fold lower than that for MTHFD2[5]. As well as in the embryo, MTHFD2 is also induced in multiple tumors to support cell proliferation via meeting the high biosynthetic demand[6]. However, whether MTHFD2 contributes to cell signaling which controls cellular activities remains to be clarified.

Resisting T cell cytotoxicity is essential for tumor development[7,8]. One paradigm involves the expression of the immunosuppressive protein PD-L1, which can directly target and activate the T cell "brake" PD-1[9]. A variety of monoclonal antibodies against PD-1 or PD-L1, such as Opdivo and Atezolizumab, have been successfully developed as promising immunotherapy drugs in non-small cell lung cancer, pancreatic cancer, breast cancer, colorectal cancer, and other solid tumors[10–12]. However, as their clinical treatment progresses, compensatory up-regulation of PD-L1 occurs in some patients, which gradually causes drug resistance[13]. Therefore, understanding the mechanism of tumor-specific up-regulation of PD-L1 and developing strategies to disrupt it are necessary.

O-GlcNAcylation, which is catalyzed at protein Ser/Thr residues by O-GlcNAc transferase (OGT)[11,14], is a fundamental way for cells to sense the availability of nutrients (glucose and glutamine)[15], and switches on multiple signals, especially for cell growth and proliferation. Moreover, basal O-GlcNAcylation is essential for cell survival, and augmentation of this modification is required for tumor development[16–18]. However, whether nutrient sources other than glucose/glutamine can trigger this signal needs further clarification.

Through functional screening, we found that folate cycle enzymes, especially MTHFD2, enable tumor cells to become more tolerant to effector T cells. MTHFD2 was overexpressed in various cancer cells and further induced by IFN-γ, which promotes both basal and IFN-γ-induced PD-L1 expression. In mechanistic terms, MTHFD2 drives the folate cycle to sustain cellular UDP-GlcNAc and cMYC O-GlcNAcylation, which enhances PD-L1 transcription. Therefore, this study uncover a role of MTHFD2 in tumor immune evasion, which elicits a powerful target in tumor immunotherapy.

## Results

**Folate cycle enzymes contribute to tumor resistance against cytotoxic T cells.** To investigate whether and how metabolic remodeling in tumor cells contributes to their resistance against T cell cytotoxicity, we sought to identify metabolic genes meeting two criteria: first, that they are upregulated during tumourigenesis; second, that they rescue tumor cells from effector T cells. Therefore, a CRISPR-based functional screen was performed. Human metastatic pancreatic cancer SW1990 cells were transduced with a sgRNA library consisting of all metabolic genes[19]. Puromycin-selected knockout (KO) cells were further treated with activated CD8+ T cells from healthy human blood. After gDNA deep-sequencing, the abundance of each residual sgRNA was compared to that of its parallel control, in order to eliminate the effect of these sgRNAs on cell viability (Fig. 1a, left). Finally, the Z-scores of 3261 genes were evaluated by MAGeCK, and the sgRNAs of 72 genes were highly (Z-score < −3) decreased after T cell treatment (Fig. 1a, right and Supplementary Data 1), suggesting that they are probably required for resisting T cell cytotoxicity. To correlate this functional screen with tumor development, the upregulated genes in human tumors were further considered. A previous meta-analysis[6] indicated that mRNAs of 3878 genes increased (Z-score > 3), and of these 1091 increased highly (Z-score ≥ 10), among 19 different types of human tumors (Supplementary Data 2/cited data). Eventually, 19 genes upregulated in tumors, including seven that were highly upregulated, were identified that fit the preset criteria (Fig. 1b and Supplementary Fig. 1a).

More interestingly, among the top 321 upregulated genes (Z-score ≥ 15) in human tumors (Supplementary Data 2), only the *methylenetetrahydrofolate dehydrogenase 2* (*MTHFD2*) was identified in this metabolic CRISPR screen (Fig. 1b and Supplementary Fig. 1b). Therefore, the functional verification of MTHFD2 was firstly performed. We depleted MTHFD2 by knockdown (KD) using siRNAs (Supplementary Fig. 1c), which showed no cytotoxicity by themselves (Supplementary Fig. 1d) but significantly aggravated T cell-induced cell death (Fig. 1c). As an embryo-specific folate cycle enzyme, MTHFD2 has also been reported to be expressed in multiple types of human tumors[6]. Likewise, we have detected notable MTHFD2 mRNA levels in a variety of cancer cell lines, particularly in SW1990 cells (Supplementary Fig. 1e). In addition, the activated human effector T cell induced more severe cell death in MTHFD2-KO SW1990 cells (Fig. 1d and Supplementary Fig. 1f), and an antigen-specific cytotoxicity assay between activated splenocytes from OT-1 mice and mouse pancreatic cancer OVA-Pan02 cells also showed an increased cell death in MTHFD2-KD Pan02 cells (Fig. 1e and Supplementary Fig. 1g).

Due to its absence in most adult tissues, MTHFD2 may be a safe target for cancer therapy. However, the role of MTHFD2 in many human tumors remains largely unknown. To define its clinical relevance, we queried the TCGA database and found that high MTHFD2 mRNA is significantly correlated with poor prognosis in pancreatic, kidney, lung, breast, and liver cancer patients (Fig. 1f). In addition, a clinical correlation for MTHFD2 has been reported in colorectal cancer patients[20]. We therefore analyzed T cell cytotoxicity in kidney cancer 786-O cells, lung cancer A549 cells, breast cancer MCF-7 cells, liver cancer HepG2 cells, and colon cancer HCT-116 cells using MTHFD2 siRNA (Supplementary Fig. 1h) and obtained results similar to those in SW1990 cells (Fig. 1c) for all these cancer cell lines (Fig. 1g and Supplementary Fig. 1i). These results suggest that at least one of

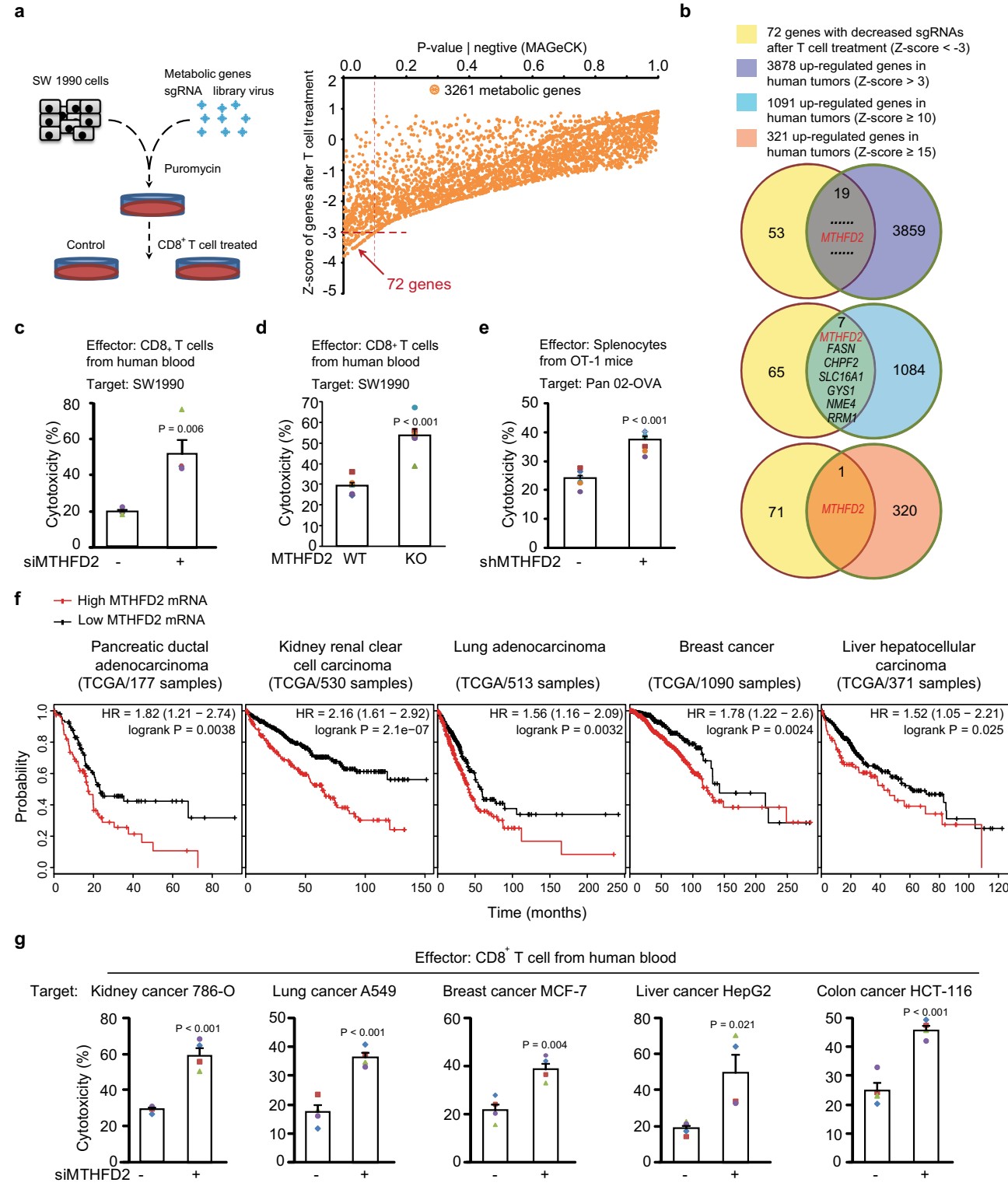

the roles for MTHFD2 in human tumourigenesis is to promote tumor immune evasion.

**MTHFD2 promotes tumor immune evasion via upregulating PD-L1 expression.** To explore the mechanism for the role of MTHFD2 in immune evasion, we analyzed the transcriptome in MTHFD2-KD SW1990 cells by RNA-seq. GSEA analysis[21] in the MSigDB C4 (cancer modules) database showed that the geneset of "immune response" was enriched in KD cancer cells (Fig. 2a),

suggesting a regulatory effect of MTHFD2 for cancer cells in response to the immune system. We searched for genes, whose mRNAs are correlated with MTHFD2 not only in cancer cells from RNA-seq (Supplementary Data 3) but also in pancreatic cancer clinical samples from the TCGA database (Supplementary Data 4). Interestingly, the well-known immune checkpoint ligand PD-L1 was found to be positively correlated with MTHFD2 in both analyses (Fig. 2b and Supplementary Fig. 2a).

Furthermore, PD-L1 mRNA (Fig. 2c) and protein (Fig. 2d, e) were decreased in MTHFD2 KD and KO human cancer cells, as

**Fig. 1 Folate cycle enzymes contribute to tumor resistance against cytotoxic T cells. a** Left: a cartoon showing the process of the CRISPR/Cas9 sgRNA library-based functional screen of metabolic genes that save SW1990 cells from human CD8[+] effector T cells; right: The Z-scores analysis of the residual sgRNA after functional screen by MAGeCK (also see Supplementary Data 1). **b** The indication of the combined analysis of the functional screen with previous meta-analysis of tumor overexpressed genes (also see Supplementary Data 1 and 2). **c–d** SW1990 cells transfected with control or MTHFD2 siRNAs (**c**) ($n = 4$ independent experiments) or sgRNAs (**d**) ($n = 6$ independent experiments) and incubated with activated human CD8[+] effector T cells for 16 h, the cytotoxicity was measured by lactate dehydrogenase (LDH) release assay. **e** OVA-Pan02 cells transfected with indicated shRNAs and incubated with activated splenocytes from OT-1 mice for 16 h, the cytotoxicity was measured by LDH release assay ($n = 7$ independent experiments). **f** Kaplan–Meier analysis of overall survival for Pancreatic ductal adenocarcinoma, Kidney renal clear cell carcinoma, Lung adenocarcinoma, Breast cancer, Liver hepatocellular carcinoma patients with low versus high expression of MTHFD2 (Kaplan–Meier analysis with the log-rank test showing in the figure from 177, 530, 513, 1090, 371 samples in the TCGA datasets.). **g** Indicated cancer cells transfected with indicated siRNAs and incubated with activated human CD8[+] effector T cells for 16 h, the cytotoxicity was measured by LDH release assay ($n = 4$ independent experiments). In (**c–e**) and (**g**), the values are presented as mean ± s.e.m.; p values (Student's t test, two-sided) with control or the indicated groups are presented (also see Supplementary Fig. 1).

well as in mouse embryonic fibroblasts (MEFs) (Fig. 2f). In contrast, exogenously expressed MTHFD2 promoted PD-L1 mRNA (Fig. 2g) and protein (Fig. 2h) expression, as well as PD-L1 transcriptional activity examined by a luciferase-based promoter assay (Fig. 2i).

PD-L1 functions only when it is localized to the cell membrane and interacts with PD-1[9]. The expression of PD-1 in the activated human CD8[+] T cells in vitro was verified (Supplementary Fig. 2b). To analyze the functional PD-L1, FACS (Fig. 2j) and purified extracellular-PD-1 domain binding analysis (Fig. 2k) were performed. MTHFD2 KD (Fig. 2j) or KO (Fig. 2k) reduced functional PD-L1, which could be reversed by exogenous expression of MTHFD2 or PD-L1. A T cell cytotoxicity assay based on LDH release (Fig. 2l and Supplementary Fig. 2c) or Real-Time-Cell-Index analysis (Supplementary Fig. 2d) showed aggravated cytolysis in MTHFD2 KD (Fig. 2l) or KO (Supplementary Fig. 2b, c) cancer cells, which was significantly reversed by the exogenously expressed PD-L1. In addition, T cell-induced cell death was aggravated by either MTHFD2 or PD-L1 depletion but only with a very mild additional effect when both were simultaneously depleted (Fig. 2m and Supplementary Fig. 2e). Accordingly, overexpressed MTHFD2 shows a compromised effect in cancer cells with PD-L1 depletion (Fig. 2n). These data suggested that MTHFD2 protects tumor cells from effector T cell cytotoxicity mainly through PD-L1. Collectively, MTHFD2 promotes functional PD-L1 at least partially through upregulating its transcriptional activity; and this role of MTHFD2 is necessary for tumor cells to evade effector T cell cytotoxicity.

**MTHFD2 is induced by IFN-γ and involved in IFN-γ-mediated PD-L1 regulation**. PD-L1 was abundant in SW1990 but not in several other cancer cell lines (Supplementary Fig. 3a). However, within the tumor mass in vivo, PD-L1 is usually maintained at a notable level, mainly due to the cytokines, especially IFN-γ, that are secreted by infiltrating lymphocytes[22]. To determine whether MTHFD2 is also involved in IFN-γ-stimulated PD-L1 expression, we examined its mRNA and protein levels in 786-O, A549, MCF-7, HepG2, and HCT-116 cells. Not surprisingly, as with basal PD-L1, IFN-γ-induced PD-L1 mRNA (Fig. 3a) and protein (Fig. 3b) were suppressed by MTHFD2-KD. Unexpectedly, meanwhile, MTHFD2 was dramatically promoted by IFN-γ in all these cancer cells (Fig. 3b) or moderately elevated in SW1990 (Supplementary Fig. 3b), probably due to the high basal level (Supplementary Fig. 1e).

To explore the underlying mechanism involved in IFN-γ induced MTHFD2, we analyzed several canonical IFN-γ downstream pathways such as AKT and STAT1. We found AKT inhibitor LY294002 (Fig. 3c and Supplementary Fig. 3b), but not STAT1 siRNA (Supplementary Fig. 3c) or inhibitor Fludara (Supplementary Fig. 3d), significantly downregulated IFN-γ-

stimulated MTHFD2. It was clear that STAT1 is critical for IFN-γ-stimulated PD-L1 but not for MTHFD2 (Supplementary Fig. 3c, d). To further analyze the classical downstream of AKT, we found the regulatory effect of IFN-γ on MTHFD2 was abolished by mTORC1 inhibitor rapamycin (Fig. 3d and Supplementary Fig. 3e), which is in line with the previous report that mTORC1 induces nucleotide synthesis via MTHFD2[23]. Rapamycin not only obviously inhibited the IFN-γ-induced PD-L1 and MTHFD2 in control cells but also slightly decreased PD-L1 in MTHFD2 KD cells (Supplementary Fig. 3f), suggesting mTORC1 induces PD-L1 obviously and partially through MTHFD2. However, p-STAT1 has not been notably affected by MTHFD2 KD (Supplementary Fig. 3f), suggesting STAT1 was neither the major upstream nor the major downstream of MTHFD2 in IFN-γ signaling. In addition, IFN-γ still displayed certain effect on PD-L1 induction in STAT1 KD cells (Supplementary Fig. 3c, g) or cells treated with STAT1 inhibitor Fludara (Supplementary Fig. 3h) but very weak or undetectable effect in cells with both STAT1 and MTHFD2 inhibition (Supplementary Fig. 3g, h), which also suggested STAT1 and MTHFD2 could independently contribute to PD-L1 expression.

MTHFD2 KD suppressed both basal and IFN-γ-elevated PD-L1 (Fig. 3a, b), indicating high basal level of tumor MTHFD2 plus IFN-γ-elevated MTHFD2 both contribute to PD-L1 expression during IFN-γ stimulation. However, IFN-γ induced notable PD-L1 expression within 4h, but MTHFD2 required 12–24 h (Fig. 3e and Supplementary Fig. 3i). This suggests that tumor basal MTHFD2 but not IFN-γ-elevated MTHFD2 is involved in the initial process of IFN-γ-induced PD-L1 expression. Collectively, these results indicated that IFN-γ promotes MTHFD2 through the AKT–mTORC1 pathway; conversely, MTHFD2 is one of the contributors for IFN-γ-stimulated PD-L1 expression (Supplementary Fig. 3j).

**Metabolic function of MTHFD2 promotes protein PD-L1 transcription through uridine-related-metabolites**. To investigate whether the metabolic role of MTHFD2 is involved in this process, the catalytically inactive MTHFD2-D168E mutant was expressed in KO cells or normal pancreatic duct epithelial (HPDE) cells. The mutant failed to up-regulate PD-L1 mRNA (Fig. 4a) and protein (Fig. 4b) as much as wild type, suggesting that its catalytic activity is an important contributor to PD-L1 expression and implying that the folate cycle pathway is involved in PD-L1 transcription.

To test this hypothesis, we performed metabolomic analysis by mass spectrometry. In agreement with the previously identified role of folate cycle, in MTHFD2-depleted cancer cells, a substantial portion of altered metabolites were nucleosides or nucleoside derivatives; moreover, serine accumulated and SAM decreased (Fig. 4c and Supplementary Data 5). Therefore, the

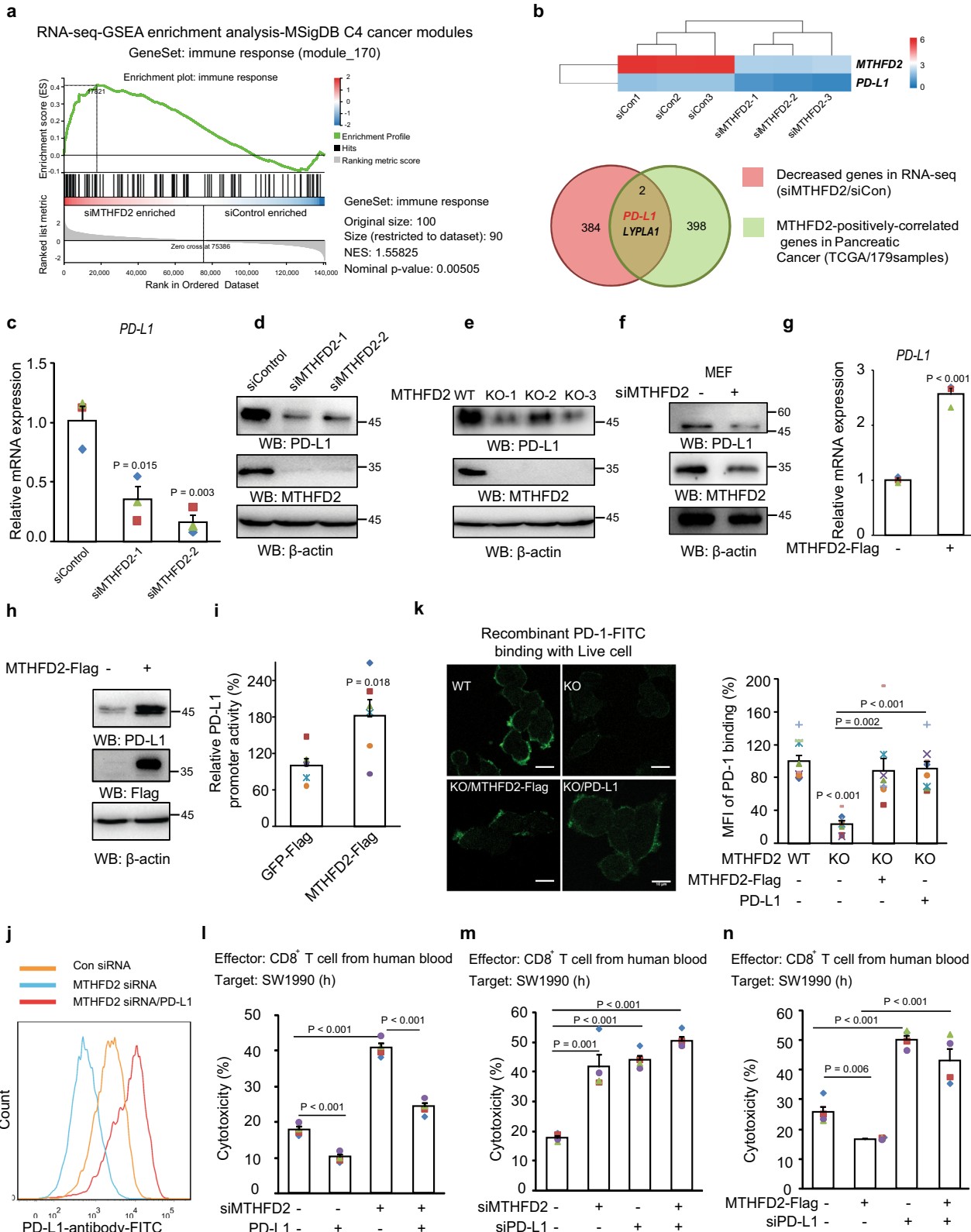

effect of these metabolites on PD-L1 was screened. It should be noted that, during exogenous treatment with these metabolites, their membrane permeability and increased intracellular concentration were not examined; and so, we cannot fully rule out metabolites with no discernable effect; on the other hand those with a detectable effect are clearly biologically relevant. More than 30 metabolites were examined, and the results for known cell-

permeable nucleosides or nucleoside derivatives with a wide range of cellular functions, such as SAM and NAM, are shown in Fig. 4d, e and Supplementary Fig. 4a–f. The second most decreased metabolite, UDP, and its related metabolite UTP significantly upregulated PD-L1 protein (Fig. 4d, e) as well as mRNA (Fig. 4f, g) in a dose-dependent manner. Uridine also promoted PD-L1, albeit to a lesser extent (Supplementary Fig. 4a).

**Fig. 2 MTHFD2 promotes tumor immune evasion via upregulating PD-L1 expression. a** The GSEA analysis of the transcriptome (RNA-seq) in SW1990 cells transfected with indicated siRNA, were performed in MSigDB C4 (cancer modules) database with the GeneSet of "immune response (module 170)". **b** The combined analysis of the genes is positively correlated with MTHFD2 mRNA both in the RNA-seq data (also see Supplementary Data 3) and in Pancreatic cancer clinical samples from the TCGA database (also see Supplementary Data 4). **c–d** SW1990 cells transfected with indicated siRNAs for 48 h, PD-L1 mRNA levels were analyzed by real-time PCR (**c**) ($n = 3$ independent experiments) and immunoblotting analyses were performed using the indicated antibodies (**d**). **e** SW1990 cells stably transduced with indicated sgRNAs by lentivirus, immunoblotting analyses were performed using the indicated antibodies. **f** MEF cells transfected with indicated siRNAs for 48 h, and immunoblotting analyses were performed using the indicated antibodies. **g–i** SW1990 cells transfected with indicated expressing vectors, PD-L1 mRNA levels were analyzed by real-time PCR (**g**) ($n = 3$ independent experiments); immunoblotting analyses were performed using the indicated antibodies (**h**); and the transcriptional activity of PD-L1 promoter (from-2000 bp to 0 bp) is examined by a luciferase-based reporter assay (**i**) ($n = 6$ independent experiments). **j** The cell-membrane localized PD-L1 in SW1990 cells with indicated treatment were analyzed by Flow cytometry using PD-L1 antibody (also see Supplementary Fig. 7a). **k** WT or MTHFD2-KO SW1990 cells transfected with indicated expressing vectors, were incubated with purified FITC-labled-recombinant-PD-1-extracellular-domain for 15 min and the cell surface localized PD-1-FITC were detected by confocal microscopy (left), scale bars: 10 μm; the FITC signal normalized to cell number were quantified ($n = 8$ independent experiments). **l–n** SW1990 cells transfected with indicated siRNAs and expressing vectors, were incubated with activated human CD8[+] effector T cells for 16 h and the cytotoxicity was measured by LDH release assay ($n = 4$ independent experiments). In (**c**, **g**, **i**, and **k–n**), the values are presented as mean ± s.e.m.; $p$ values (Student's $t$ test, two-sided) with control or the indicated groups are presented (also see Supplementary Fig. 2).

Additionally, both UTP and UDP compromised the inhibitory effect of MTHFD2-KD on PD-L1 mRNA (Fig. 4h) and promoter activity (Fig. 4i). These results suggest that MTHFD2 promotes PD-L1 transcription through enhancing cellular UTP/UDP.

**Protein O-GlcNAcylation mediates MTHFD2 and uridine-related-metabolites enhanced PD-L1 transcription.** To investigate how UTP/UDP mediates this process, we further analyzed all the uridine-related metabolites (Fig. 5a, upper). Strikingly, UDP-GlcNAc, which is synthesized from UTP via the hexosamine biosynthesis pathway (HBP) and functions as a substrate for protein O-GlcNAcylation[14] (Fig. 5a, lower), was one of the most highly decreased metabolites in KD cells. We therefore, further examined the cellular protein O-GlcNAcylation, which was significantly suppressed in KD cells and recovered after exogenous UTP/UDP treatment (Fig. 5b). Moreover, the canonical O-GlcNAcylation induction by high glucose was also compromised in MTHFD2 KD cells, suggesting both glucose and nucleotide promoted by MTHFD2 could be the limiting metabolite for O-GlcNAc modification (Supplementary Fig. 5a). Enhanced O-GlcNAcylation resulting from overexpressed OGT not only promoted basal PD-L1 protein (Supplementary Fig. 5b) and mRNA (Supplementary Fig. 5c), but also rescued MTHFD2-KD-suppressed O-GlcNAcylation, PD-L1 protein (Fig. 5c) and mRNA (Fig. 5d) as well as PD-L1 promoter activity (Fig. 5e). The PD-L1 protein as well as global O-GlcNAcylation could be obviously enhanced by overexpressing MTHFD2, and this effect was abrogated by OGT KD (Fig. 5f). In addition, T cell cytotoxicity assay indicated aggravated cytolysis in MTHFD2 KD cells was significantly reversed by the exogenously expressed OGT (Fig. 5g).

Protein O-GlcNAcylation plays fundamental roles in various cellular activities. Having uncovered its relationship with MTHFD2 and PD-L1 in cultured cancer cells, we next wished to confirm the clinical relevance of these associations. IHC was performed to semi-quantify the MTHFD2, O-GlcNAc, and PD-L1 levels in 73 consecutive human pancreatic tumor specimens (Fig. 5f). The quantified global O-GlcNAc levels were consistently, highly, and positively correlated with MTHFD2 (Figs 5f, g). Meanwhile, the PD-L1 levels were also positively correlated with global O-GlcNAc (Fig. 5f, h). These data support the clinical relevance of our finding that protein O-GlcNAcylation mediates MTHFD2-enhanced PD-L1 expression.

**MTHFD2 promotes PD-L1 transcription through cMYC O-GlcNAcylation.** Many cellular proteins are functionally affected

by O-GlcNAc modification. To identify which O-GlcNAcylated protein(s) contribute to PD-L1 transcription, we further analyzed transcription factors, which might be functionally affected by MTHFD2-KD, in a transcriptome study (Fig. 2a). GSEA analysis was performed in database MSigDB C6 (oncogenic signatures) and MSigDB C3 (transcription factor targets)[21]. Interestingly, the transcription factor cMYC targeted GeneSets in both databases that were enriched in control but not KD cancer cells (Fig. 6a). This suggests that the transcriptional activity of cMYC is suppressed in MTHFD2-KD cancer cells. Moreover, a previous report has shown that cMYC can directly target the PD-L1 promoter[24]. In support of that observation, cMYC siRNA suppressed the PD-L1 protein level (Supplementary Fig. 6a), while exogenously expressed cMYC elevated it (Supplementary Fig. 6b). cMYC-KD decreased the transcriptional activity of the PD-L1 promoter, which could be rescued by overexpressed cMYC (Supplementary Fig. 6c). Furthermore, overexpressed cMYC dramatically rescued MTHFD2-KD-suppressed PD-L1 mRNA (Fig. 6b). However, MTHFD2-KD did not affect cMYC mRNA (Supplementary Fig. 6d), but obviously suppressed cMYC and PD-L1 protein levels, which could be recovered by both exogenous UTP and UDP (Fig. 6c). These data suggest that cMYC acts downstream of MTHFD2/UTP/UDP and contributes to PD-L1 transcription. However, whether the O-GlcNAcylation of cMYC is involved in this process remained to be clarified.

Interestingly, previous mass spectrometry has identified cMYC Thr58 as being O-GlcNAcylated[25], but the biological role of this modification remains to be explored. Phosphorylation of Thr58, however, has been reported to be a key event for cMYC ubiquitination and degradation[26,27]. Thus, it is conceivable that enhanced cMYC O-GlcNAcylation precludes phosphorylation at the same residue and thereby blocks cMYC protein degradation. To test that, the mutant cMYC-T58A, in which both modifications are disrupted, was exogenously expressed in cancer cells. Consistently, T58A mutant cMYC protein exhibited longer half-life time than the wild type (Fig. 6d). The enhanced O-GlcNAcylation by overexpressed OGT obviously stabilized wild type but not mutant cMYC protein (Fig. 6d and Supplementary Fig. 6e), which proved the hypothesis that O-GlcNAcylation at Thr58 promotes cMYC protein stability. The Flag-cMYC pull-down and O-GlcNAcylation analyses were further performed. Indeed, WT but not mutant cMYC showed a notable level of O-GlcNAcylation, which could be further enhanced by over-expressed MTHFD2 (Fig. 6e, upper) or suppressed by MTHFD2-KD (Fig. 6f, upper). This observation indicates that Thr58 is the major MTHFD2-enhanced cMYC O-GlcNAcylation site. Moreover, overexpressed MTHFD2 obviously increased

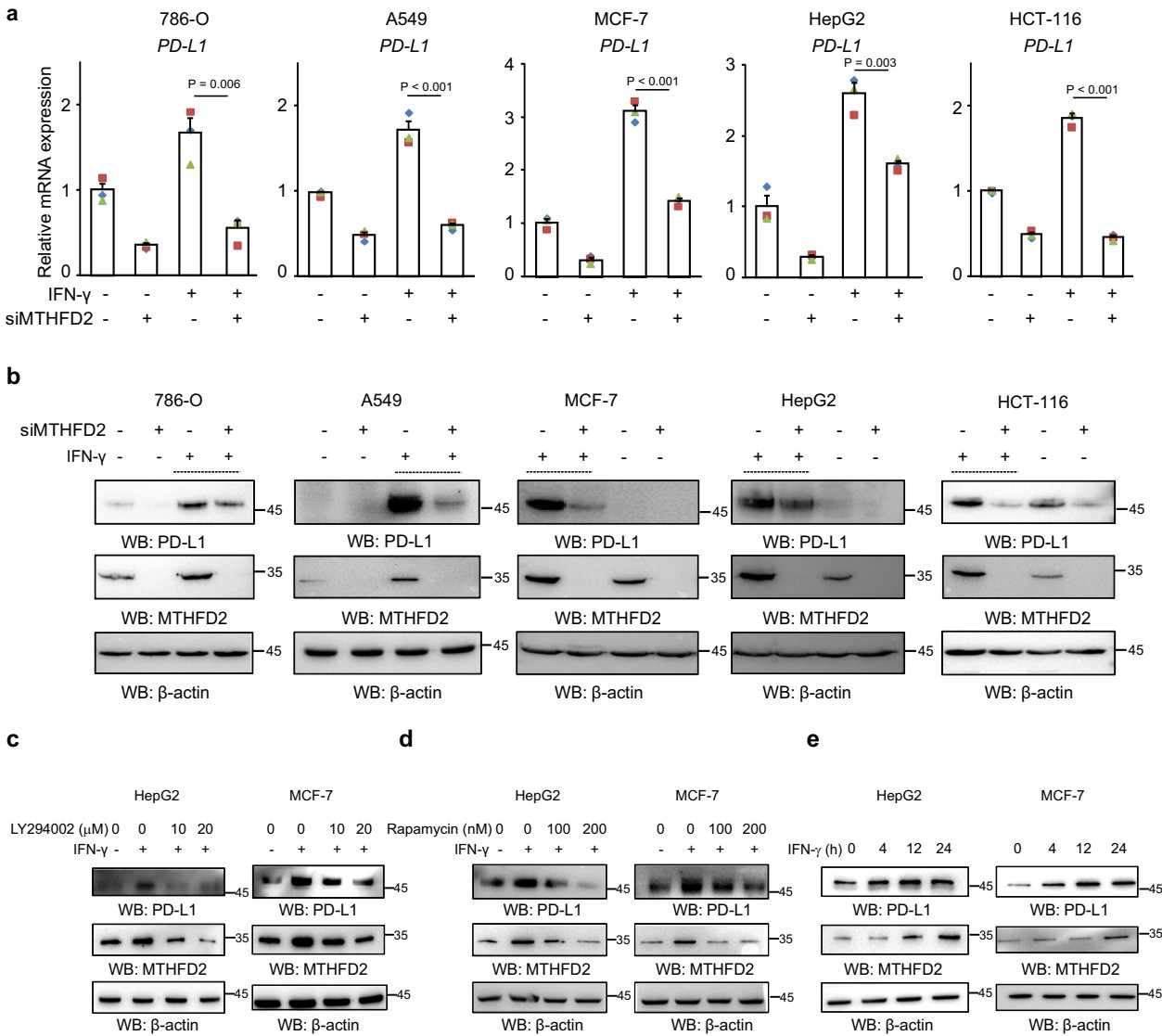

**Fig. 3 MTHFD2 is induced by IFN-γ and involved in IFN-γ-mediated PD-L1 regulation. a**, **b** Indicated cancer cells transfected with control or MTHFD2 siRNAs for 24 h and then treated with 0 or 20 ng/ml IFN-γ for another 24 h, PD-L1 mRNA levels were analyzed by real-time PCR (**a**) and immunoblotting analyses were performed using the indicated antibodies (**b**). **c**, **d** Indicated cancer cells pretreated with LY294002 (**c**) and Rapamycin (**d**) with indicated concentrations and then cultured with 20 ng/ml IFN-γ, immunoblotting analyses were performed using the indicated antibodies. **e** Indicated cancer cells treated with 20 ng/ml IFN-γ for indicated times, immunoblotting analyses were performed using the indicated antibodies. In (**a**), the values are presented as mean ± s.e.m (n = 3 independent experiments); p values (Student's t test, two-sided) with control or the indicated groups are presented (also see Supplementary Fig. 3).

(Fig. 6e, lower) while MTHFD2-KD decreased the total cMYC protein (input) (Fig. 6f, lower), but affected cMYC-T58A relatively mildly, which suggests that O-GlcNAcylation at Thr58 is a major reason for MTHFD2-maintained cMYC protein level (Fig. 6e, f). In addition, PD-L1 mRNA (Fig. 6g) and protein (Fig. 6h), as well as both O-GlcNAcylated and total cMYC proteins (Fig. 6h), were suppressed by MTHFD2-KD and recovered by exogenous UTP/UDP in cMYC-overexpressing cancer cells, all of which were compromised in MYC-T58A-overexpressing cells. Taken together, these results indicate that the MTHFD2-dirven folate cycle sustains sufficient cellular UDP-GlcNAc, which in turn promotes MYC protein stability by O-GlcNAcylation at Thr58, resulting in PD-L1 transcription.

**Upregulation of PD-L1 by MTHFD2 is required for tumorigenesis.** To further explore the role of MTHFD2 in tumor development in vivo, mouse pancreatic cancer Pan02 cells were

subcutaneously injected into both athymic nude mice and C57 mice. The shRNA targeting mouse MTHFD2 suppressed tumor growth moderately in nude mice (Fig. 7a), but dramatically (around 80%) in C57 mice, which contain mature lymphocytes (Fig. 7b). These results suggest that MTHFD2, which is required to maintain PD-L1 mRNA (Fig. 7c) and protein (Fig. 7d) in the tumor mass, promotes tumourigenesis largely by undermining lymphocytes. Moreover, the inhibitory effects of shMTHFD2 on tumor growth (Fig. 7b) and PD-L1 expression (Fig. 7c, d) in C57 mice were dramatically reversed by the exogenous overexpression of PD-L1. Meanwhile, shMTHFD2 facilitated the infiltration of CD8[+] T cells (Fig. 7e) as well as total CD45 immune cells (Fig. 7f) into tumor mass, both of which were largely negated by exogenously expressed PD-L1 (Fig. 7e, f). Taken together, these data indicate that the PD-L1-up-regulatory effect of MTHFD2 is necessary for tumor development.

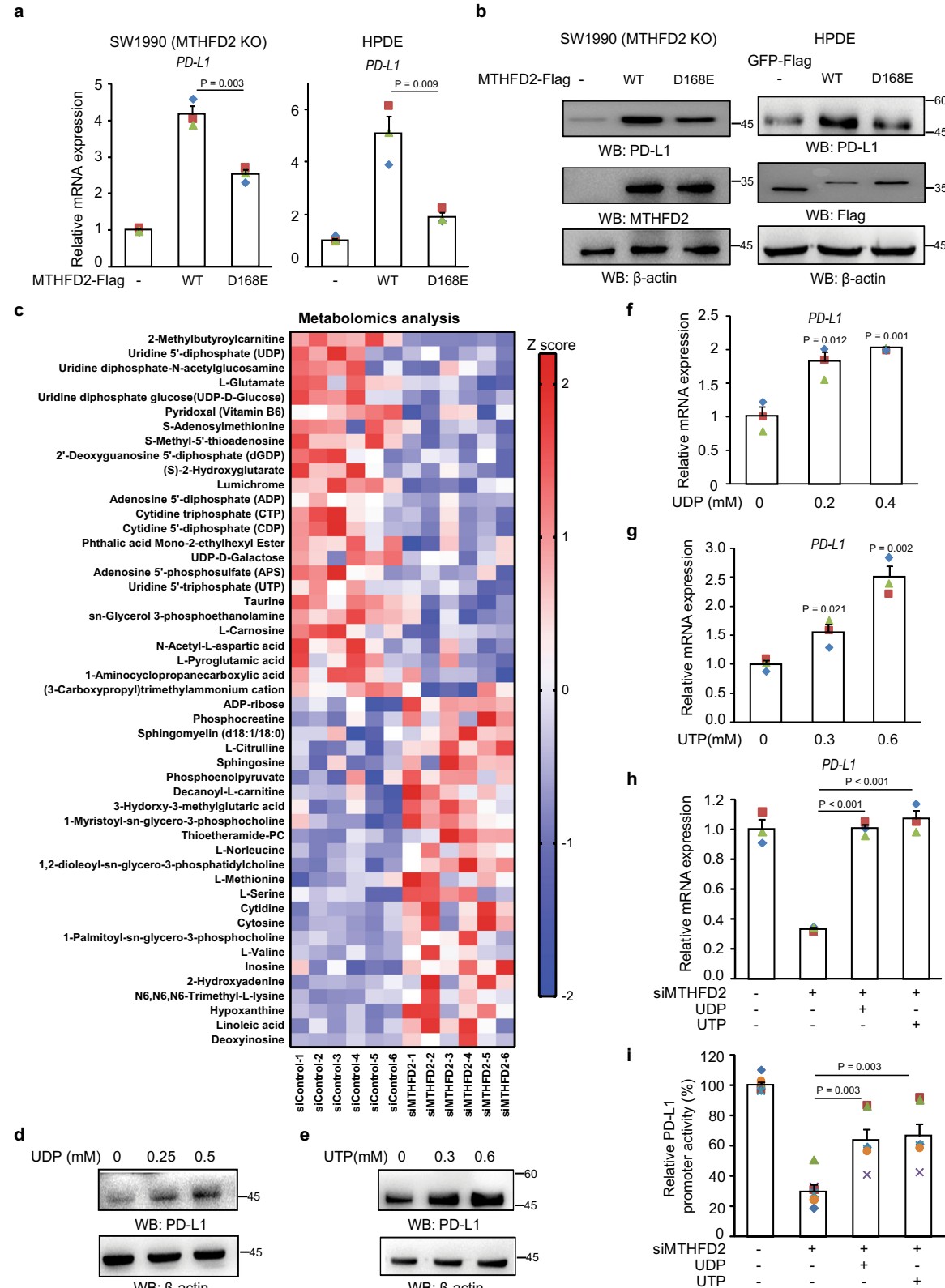

## Discussion

Tumor cells have to adapt to changing microenvironments, which contain not only nutrients but also immune cells. Reprogramming metabolic pathways to meet cellular bioenergetic and biosynthetic demands happens in both cancer and immune cells within the tumor mass[28,29], and may affect the confrontation between them. Previous studies focus on the affection of immune cell signaling by the metabolic alteration no matter what happens in immune cells or tumor cells[29–31]. In this study, we screened the corresponding functional role of metabolic genes from the perspective of tumor cells, and demonstrated that MTHFD2, an embryo- and tumor-specific folate cycle enzyme, could be

**Fig. 4 Metabolic function of MTHFD2 promotes protein PD-L1 transcription through uridine-related-metabolites. a–b**, MTHFD2-KO SW1990 cells and HPDE cells transfected with indicated expressing vectors, PD-L1 mRNA levels were analyzed by real-time PCR (**a**) and immunoblotting analyses were performed using the indicated antibodies (**b**). **c** The metabolomics analysis in SW1990 cells with siMTHFD2 (n = 6 independent samples). **d–g** SW1990 cells were treated with indicated concentration of indicated metabolites for 24 h, immunoblotting analyses were performed using the indicated antibodies (**d**, **e**) and PD-L1 mRNA levels were analyzed by real-time PCR (**f**, **g**). **h**, **i** SW1990 cells with siMTHFD2 were treated with 0.3 mM UDP or 0.6 mM UTP for 24 h, PD-L1 mRNA levels were analyzed by real-time PCR (**h**) and the transcriptional activity of PD-L1 promoter (from 2000 bp to 0 bp) is examined by a luciferase-based reporter assay (**i**). In (**a**) and (**f–i**), the values are presented as mean ± s.e.m (n = 3 independent experiments); p values (Student's t test, two-sided) with control or the indicated groups are presented (also see Supplementary Fig. 4).

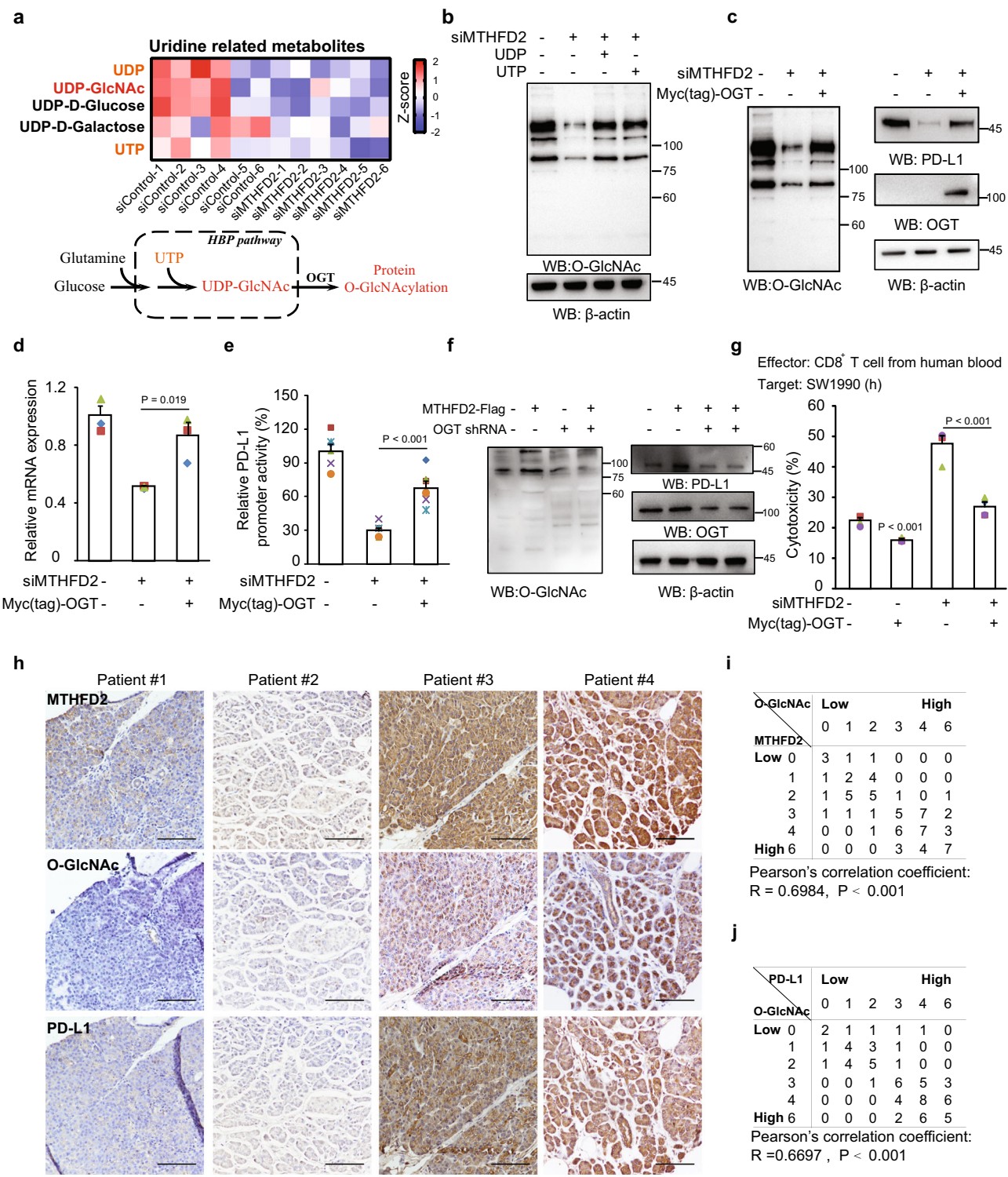

**Fig. 5 Protein O-GlcNAcylation mediates MTHFD2 and uridine-related-metabolites enhanced PD-L1 transcription. a** The Uridine-related metabolites analysis in SW1990 cells with siMTHFD2 (upper). Diagram for protein O-GlcNAcylation modulated by metabolites in the hexosamine biosynthetic pathway (lower). **b** SW1990 cells with siMTHFD2 were treated with 0.3 mM UDP or 0.6 mM UTP for 24 h, and immunoblotting analyses were performed using the indicated antibodies. **c–e** SW1990 cells with siMTHFD2 were expressed with Myc-OGT, and immunoblotting analyses were performed using the indicated antibodies (**c**), PD-L1 mRNA levels were analyzed by real-time PCR (**d**) ($n = 3$ independent experiments) and the transcriptional activity of PD-L1 promoter (from 000 bp to 0 bp) was examined by a luciferase-based reporter assay (**e**) ($n = 6$ independent experiments). **f** SW1990 cells transfected with indicated expressing vectors, immunoblotting analyses were performed using the indicated antibodies. **g** SW1990 cells transfected with indicated siRNAs and expressing vectors, were incubated with activated human CD8$^+$ effector T cells for 16 h and the cytotoxicity was measured by LDH release assay ($n = 4$ independent experiments). **h** Immunohistochemical staining with indicated antibodies was performed on 73 consecutive human pancreas tumor specimens. Representative images are shown. Scale bars: 100 μm. **i** Semi-quantitative scoring between MTHFD2 and O-GlcNAc were performed (Pearson's correlation test; $R = 0.6986$, $P < 0.001$.). **j** Semi-quantitative scoring between O-GlcNAc and PD-L1 were performed (Pearson's correlation test; $R = 0.6697$, $P < 0.001$.). In (**d**, **e**, and **g**), the values are presented as mean ± s.e.m.; $p$ values (Student's $t$ test, two-sided) with control or the indicated groups are presented (also see Supplementary Fig. 5).

induced by IFN-γ. Meanwhile, MTHFD2 promotes PD-L1-mediated tumor immunoresistance through the folate cycle-HBP metabolic pathway and the UDP-GlcNAc-O-GlcNAcylation-MYC-PD-L1 signaling pathway.

In this study, we focused only on the PD-L1 expression to explore the functional role of MTHFD2 in tumor immunity, however, we have shown unequivocally that MTHFD2 plays an important role in global protein O-GlcNAcylation, which is known to fundamentally regulate various cellular behaviors. According to its expression pattern, the physiological relevance of MTHFD2 limits in embryonic or immune T cells, in which this particular mechanism was tested in this study but more are need. Our findings suggest that more functional roles for MTHFD2 in cell biology remain to be clarified.

IFN-γ, which was first identified as an anti-tumor cytokine, can be secreted by activated lymphocytes[32,33]. However, IFN-γ clearly stimulates tumor progression in some clinical trials[34,35]. Induction of the onco-metabolic enzyme MTHFD2, which has been shown here to promote tumor development in both immune-dependent and -independent ways, may be a major cause of the pro-tumor effect of IFN-γ in cancer treatment. Therefore, combining IFN-γ with a strategy to inhibit tumor-expressed MTHFD2 may offer a better approach to cancer immunotherapy. Moreover, our study also shows a supportive role for MTHFD2 in immunity. These suggest that, beyond cancer, MTHFD2 may play more roles in immune biology, given the diverse functions of MTHFD2 in cellular activities.

Cell growth means the increase of cellular components, mainly proteins, lipids, and nucleotides, of whose production is centrally controlled by mTORC1 signaling. mTORC1 promotes protein synthesis through S6K1/4EBP1 and lipid synthesis through SREBP. Interestingly, mTORC1 induces nucleotide synthesis through MTHFD2[23], which itself sustains notable mTORC1 activity and a high protein synthetic according to our study. This mutual enhancement relationship between mTORC1 and MTHFD2 may play more widespread roles in anabolism and other cellular activities.

Incubation of cancer cells with exogenous UTP/UDP has been reported to promote cell proliferation[36–38]. Here, we show that these uridine-related metabolites determine cellular O-GlcNAcylation levels, which are required for maintaining multiple cell proliferation pathways. Accordingly, cellular O-GlcNAcylation should be considered as a nutrient sensor not only for carbohydrates but also for nucleotides.

Some metabolic enzymes, such as FBP1[39] and PKM2[40], have been reported to control metabolism while regulating signaling pathways, as a result of which metabolic status is coordinated with cell fate. Here, we have shown another important paradigm. MTHFD2 maintains a high level of cellular nucleotides, while stimulating signaling for cell proliferation accompanied by higher nucleotide consumption.

Since metabolism is essential to support almost all cellular activities, it is not surprising that numerous metabolic enzymes have been reported to be involved in tumor development with different mechanisms. However, due to their fundamental roles in cell biology, targeting these enzymes for cancer therapy could bring unexpected adverse effects. In this context, MTHFD2, which is rarely expressed in normal adult tissues, where it has a functional substitute in MTHFD2L, may well be a safe therapeutic target for cancer treatment.

## Methods

**Materials**. Antibodies that recognize MTHFD2 (D8W9U, no.41377,1:1,000), PD-L1 (E1L3N®, no.13684, 1:1,000), O-GlcNAc (CTD110.6, no. 9875, 1:1,000) p-STAT-1 T701 (58D6, no.9167, 1:1,000), STAT-1 (D1K9Y, no.14994, 1:1000) were purchased from Cell Signaling Technology. OGT (ab96718, 1:1,000) and PD-L1 (ab205921, 1:1,000) were purchased from Abcam. Anti-FLAG® M2 Magnetic Beads (M8823) and β-Actin−Peroxidase antibody (A3854) were purchased from Sigma. Antibodies against flag-tag (PA1-984B, 1:1000) was obtained from Invitrogen. Antibodies against Myc (10828-1-AP, 1:1,000) were purchased from Proteintech. CD45 eFluor 450 (B220, 48-0452-82) and CD8a monoclonal antibody (4SM15, 14-0808-82) were obtained from eBioscience. Human CD3 (300314), human CD28 (302914), mouse CD3 (100207), mouse CD28 (102111) activation antibodies were obtained from Biolegend. EasySep Human CD8 Pos Sel Kit (17853), Lymphoprep (07851) and EasySep$^{TM}$ (18000) were purchased from STEMCELL Technologies. IL-2 (589102) were obtained from Biolegend; human IFN-γ (11725-HNAS) and mouse IFN-γ (50709-MNAH) were obtained from Sino Biological.

siRNAs including MTHFD2, MYC, STAT1, PD-L1 and OVA (SIINFEKL) peptide were synthesized by Synbio technologies. siRNA sequences are listed as below:

siControl: UUCUCCGAACGUGUCACGUdTdT
siMTHFD2-1: GGAUGCUUCACUUUGUCAAdTdT
siMTHFD2-2: UGGCAAUGCUAAUGAAGAAdTdT
sicMYC: GAAUUUCAAUCCUAGUAUAdTdT
siSTAT1-1: CUGACUUCCAUGCGGUUGAdTdT
siSTAT1-2: CGGCUGAAUUUCGGCACCUdTdT
siPD-L1: CAAAAUCAACCAAAGAAUUdTdT

LY294002 (HY-10108) was purchased from MCE. Rapamycin(abs810030) was purchased from Absin. UDP (A600979), UTP (A620570), uridine (A610570), adenine(A600013), guanine (A610246), adenosine(A600016), guanosine(A610250), cytosine(A600138), S-adenosyl-L-methionine (SAM, A506555), nicotinamide (NAM, A510659) were purchased from Sangon. Fludara (HY-B0069) and Cycloheximide (HY-B1248) were purchased from MCE.

The Research Resource Identifiers (RRIDs) for some key reagents are provided below:

pCDH-Flag-c-Myc RRID:Addgene_102626
Human CRISPR Metabolic Gene Knockout Library RRID:Addgene_110066
MTHFD2 (D8W9U) Rabbit mAb antibody,
Cell Signaling Technology Cat# 41377, RRID:AB_2799200
O-GlcNAc (CTD110.6) Mouse mAb antibody,
Cell Signaling Technology Cat# 9875, RRID:AB_10950973
PD-L1 (E1L3N) XP Rabbit Antibody,
Cell Signaling Technology Cat# 13684, RRID:AB_2687655

**Cell lines**. SW1990 cells were cultured in RPMI1640 supplemented with 10% fetal bovine serum (FBS). HPDE, HepG2, A549, MCF-7, HCT-116, 786-O and Pan02 cell lines were cultured in Dulbecco's modified Eagle's medium (DMEM) supplemented with 10% FBS. They were maintained in a cell culture incubator at 37 °C under a humidified atmosphere with 5% CO$_2$. SW1990, HepG2, A549, MCF-7,

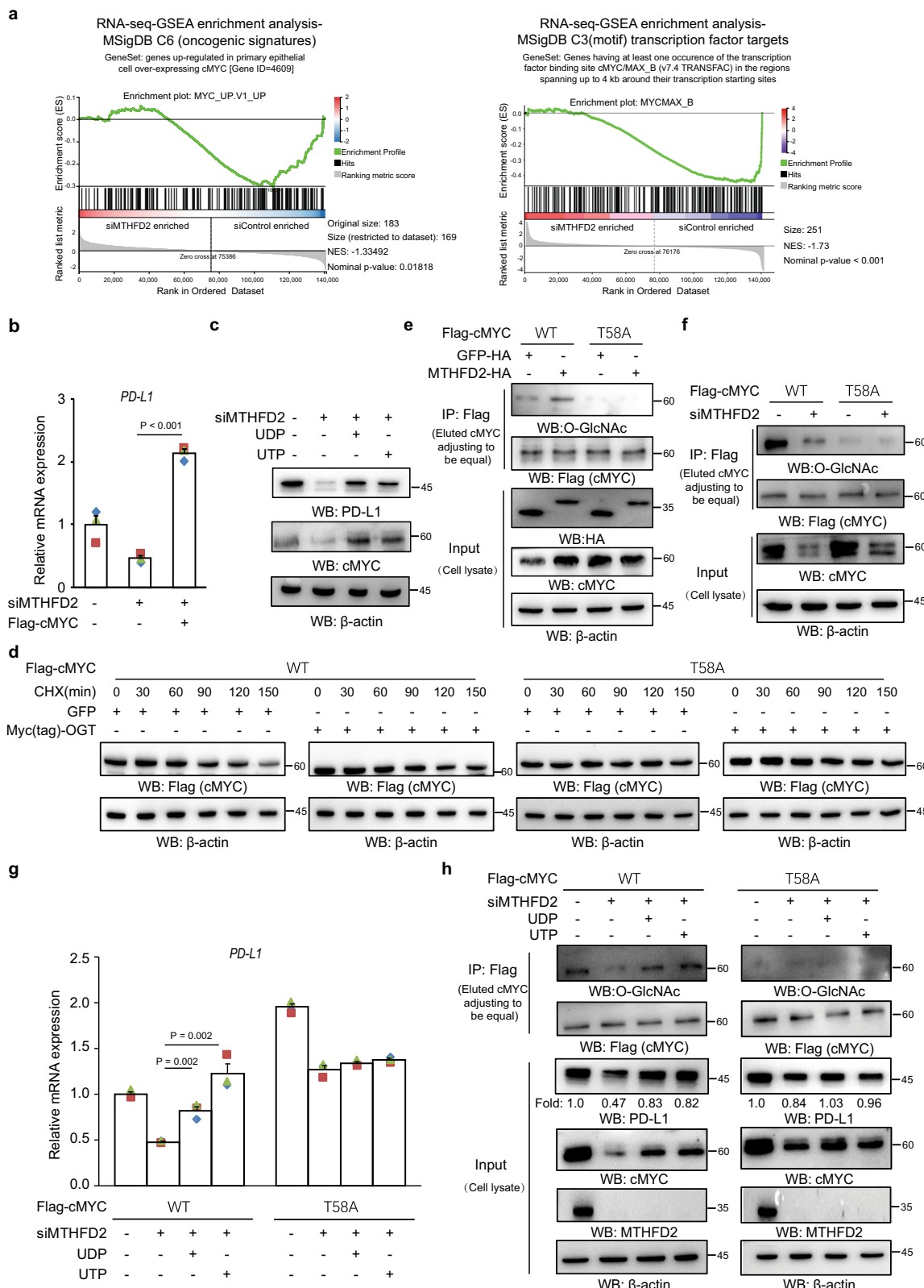

HCT-116, 786-O and Pan02 cell lines were from ATCC. HPDE cell line was from kerafast. All cell lines were confirmed to be mycoplasma free.

**DNA constructs**. The lentiviral sgMTHFD2 vectors were generated via ligation of hybridized oligos (below) into lentiCRISPR-v1 vector linearized with BsmBI (NEB) using T4 DNA ligase (NEB). Primers are shown in Supplementary Data 6.

The plenti-MTHFD2-FLAG were generated by cloning MTHFD2 cDNA ordered from DNA core in Shanghai Jiao Tong University into the plenti-GFP-FLAG vector (OBiO Technology) digested by ECORI and BamHI using Syno assembly (Synbio technologies). The plenti-MTHFD2-FLAG-D168E were cloned with mutagenic primers containing desired mutation using the QuikChange II site-directed mutagenesis kit (Agilent Technologies). Primers are shown in Supplementary Data 6.

**Fig. 6 MTHFD2 promotes PD-L1 transcription through cMYC O-GlcNAcylation. a** The GSEA analysis of the transcriptome (as indicated in Fig. 2a) in SW1990 cells transfected with indicated siRNAs, were performed with the GeneSet "Genes regulated in primary epithelial cell overexpressing cMYC" in MSigDB C6 (oncogenic signatures) and with the GeneSet "Genes having at least one occurrence of the transcription factor binding site MYC/MAX_B in the regions spanning up to 4 kb around their transcription starting sites" in MSigDB C3 (motif) transcription factor targets database. **b** SW1990 cells with siMTHFD2 transfected with indicated expressing vectors, PD-L1 mRNA levels were analyzed by real-time PCR. **c** SW1990 cells with siMTHFD2 treated with 0.3 mM UDP or 0.6 mM UTP for 24 h, immunoblotting analyses were performed using the indicated antibodies. **d** Analysis of the apparent half-life time of wild type and mutant cMYC in cycloheximide treated cells transfected with indicated vectors. **e, f** The cell lysates of SW1990 cells expressing WT or T58A Flag-cMYC with indicated plasmid (**e**) or siRNA (**f**) were subjected to immunoprecipitation with an anti-Flag antibody, and the precipitates (loading volume adjusted to be equal precipitated cMYC) and whole lysates (equal total protein amount loading) were analyzed by immunoblotting using indicated antibodies. **g, h** SW1990 cells expressed with WT or T58A Flag-cMYC transfected with indicated siRNAs, were treated with 0.3 mM UDP or 0.6 mM UTP for 24 h. PD-L1 mRNA levels were analyzed by real-time PCR (**g**); cell lysates were subjected to immunoprecipitation with an anti-Flag antibody, and the precipitates (loading volume adjusted to be equal precipitated cMYC) and whole lysates (equal total protein loading) were analyzed by immunoblotting using indicated antibodies (**h**). In (**b**) and (**g**), the values are presented as mean ± s.e.m ($n = 3$ independent experiments); $p$ values (Student's $t$ test, two-sided) with control or the indicated groups are presented (also see Supplementary Fig. 6).

---

The plenti-PD-L1-FLAG were generated by cloning an PD-L1 PCR fragment into the plenti-GFP-FLAG vector digested by EcoRI and BamHI using Syno assembly (Synbio technologies).

The mouse shMTHFD2 fragment was synthesized by Genewiz. The pGIPZ mouse MTHFD2 shRNA was generated by inserting the shMTHFD2 fragment into the pGIPZ vector.

The pcDNA3-Myc-OGT and pGIPZ human OGT shRNA was described as previously[41]. Briefly, pGIPZ human OGT shRNA was generated by inserting the shOGT fragment into the pGIPZ vector. Primers are shown in Supplementary Data 6. pCDH-Flag-Myc was a gift from Hening Lin (Addgene plasmid # 102626 ; http://n2t.net/addgene:102626 ; RRID:Addgene_102626).

The pCDH-FLAG-MYC-T58A were cloned with mutagenic primers containing desired mutation using the QuikChange II site-directed mutagenesis kit (Agilent Technologies). Primers are shown in Supplementary Data 6.

The pet28a-PD-1 was cloned by inserting PD-1 extracellular fragment synthesized by Genewiz into pet28a plasmid linearized with EcoRI (Thermofisher).

**Culture of mouse embryonic fibroblasts (MEF).** Fetuses were harvested between day 12.5-14.5d of gestation. Tissues of embryo except head and liver were washed, minced and digested with 2 ml of trypsin. Fetal tissues were vigorously pipetted until they became a single cell suspension. Cells were cultured in DMEM containing 10% FBS and split every 3–4 days.

**Isolation of peripheral blood mononuclear cells (PBMCs) and CD8+ T cells.** Buffy coats of healthy donors were purchased from Tianjin Blood Center. The blood samples were only used for the present research and approved by the Ethics Committee of Tianjin Blood Center. PBMCs were separated by density gradient centrifugation in Lymphoprep (STEMCELL). Briefly, the buffy coat was added in the conical tube pre-added with the same volume of Lymphoprep and centrifuged at 400$g$, 30 min, 4 °C with brakes off. The interphase was carefully moved into a new conical tube and diluted with three-volume of PBS, followed by centrifugation at 300$g$, 10 min Then the cell pellets were washed twice with 25–30 ml PBS by centrifugation at 200$g$, 15 min The remaining pellets were considered as PBMCs and suspended at a density of $1 \times 10^8$ cells/ml.

Isolation of CD8+ T cells from PBMCs was performed using EasySep Human CD8 Pos Sel Kit (STEMCELL) according to the manufacturer's instructions. The resuspended cells at the indicated concentration within the volume of 0.1–2.5 ml were transferred into a 5 ml polystyrene round-bottom tube. PBMCs were firstly supplemented with 100 μl selection cocktail, mixed and incubated for 3 min at room temperature (RT), then mixed with 50 μl RapidSpheres™ and incubated under the same condition. The cells were isolated using the magnet after the incubation. The isolated CD8+ T cells were resuspended in RPMI 1640 medium and incubated with 2 μg/ml CD3, CD28 and 10 ng/ml IL-2 (Biolegend) for 48–96 h. The activated CD8+ T cells were used for the following T cell killing assay.

**CD8+ T cell-mediated tumor cell killing assay.** Adherent tumor cells including SW1990, HepG2, A549, MCF-7, HCT-116, and 786O ($4 \times 10^4$ cells/well) were plated in 24-well plates. All tumor cells were transfected with indicated siRNAs or plasmids. The activated CD8+ T ($2 \times 10^5$ /well) were added to each well at an effector-to-target ratio of 5 and co-cultured with adherent tumor cells for 16 h. At the end of incubation, the plates were centrifuged at 400$g$, 5 min The supernatants in each group were collected for LDH release assay (Beyotime) according to the manufacturer's instructions. The absorbance was detected at 490 nm using a Biotek microplate reader.

**Metabolic CRISPR/Cas9 sgRNA library screen.** Human CRISPR Metabolic Gene Knockout Library was obtained from Addgene (110066). The pLentiCRISPR sgRNA library was packaged with PMD2.G envelope plasmid and psPAX2

lentiviral packaging plasmid. The transduction efficiency of SW1990 cells was maintained at 30% and approximately 300 cells could be infected with per lenti-CRISPR construct. About $3 \times 10^7$ SW1990 cells were infected with the metabolic CRISPR/Cas9 sgRNA library targeting all human metabolic genes[19]. Cells were infected for 48 h and screened with puromycin for another 72 h. After 48 h of recovery post-selection, SW1990 cells were passaged to indicated Control and T cell-killing group containing two replicates, respectively. The activated CD8+ T cells were added to T cell-killing group at an effector-to-target ratio of 2 and co-cultured with adherent tumor cells for 36 h. Both control and T cell-killing groups continued to be incubated for 72 h. Each DNA sample was extracted from $2 \times 10^7$ cells using the Blood & Cell Culture DNA Midi Kit (QIAGEN). Amplification was carried out with 18 cycles for the first PCR and 24 cycles for the second PCR. For the first PCR, sgRNA fragments were amplified using Phusion Hotstart DNA Polymerase (Thermofisher). A second PCR was performed to attach Illumina adaptors and to barcode samples. The abundance of each sgRNA in the pooled samples was determined by deep sequencing on an Illumina platform. Primers sequences to amplify lentiCRISPR sgRNAs for the first PCR are shown in Supplementary Data 6.

Sequencing reads were aligned to the sgRNA library and the abundance of each sgRNA was calculated. All the normalized sgRNA read counts with more than 5 fold-change in one group between the two samples were removed from analysis. A Z-score was calculated using the mean and s.d. of normalized read counts between Control and T cell killing samples. Z-score results were analyzed by MAGeCK. Genes with highly decreased sgRNAs after T cell treatment were screened out as the difference in p value < 0.1, z-score < −3.

**Transfections and treatments.** SW1990, HPDE, HepG2, A549, MCF-7, HCT-116, 786O and Pan02 cells were transfected with different plasmids with Lipofectamine 3000 (Thermofisher) and with siRNAs using RNAi Max (Thermofisher) according to the manufacturer's instructions.

LY294002 were used at 10, 20 μM while rapamycin at 100, 200 nM for 24 h, respectively. For IFN-γ stimulation, cells were treated with 20 ng/ml IFN-γ for 4, 8, 12, 24 h. UDP, UTP, uridine, adenine, guanine, adenosine, guanosine, cytosine, SAM, NAM (Sangon) were added to cells for 24 h at indicated concentrations. Fludara was used at 20, 50 μM for 24 h on SW1990 cells. Cycloheximide was used at 50 μg/ml for indicated time.

**Animals.** All animal studies were performed under the guidelines of the Animal Care and Use Committee of Tianjin Medical University. All animals were kept under specific pathogen free (SPF) and temperature-controlled environment with 12 h light/12 h dark cycle, and free access to food and water. C57BL/6J mice and BALB/c nude mice for tumor xenografts were purchased from Beijing Vital River Laboratory Animal Technology Co., Ltd. OT-I mice were kindly provided by the third military medical university. Six-week-old male nude mice ($n = 8$ for each group) were injected with $2 \times 10^6$ Pan02 cells in a volume of 150 μl PBS. Xenografts were planted subcutaneously in the left and right flanks, respectively. Tumor volume was calculated using formula "volume = (width$^2$) × length/2. Six-week-old male C57BL/6J mice were injected with $5 \times 10^6$ Pan02 cells for each xenograft. Tumor xenografts were collected on the 28th day. All animal experiments were approved by the Ethics Committee of Tianjin Medical University.

**Isolation of mouse splenocyte and cell killing assay.** OT-I mice were sacrificed and placed on a clean dissection board. The spleen was removed through laparotomy and pressed through a 70-μm mesh in 10 ml of PBS to disassociate the tissue. The solution containing splenocyte was then centrifuged at 400$g$ for 5 min The cell pellet was washed and resuspended in culture medium supplemented with 2 μg/ml CD3, CD28 and 10 ng/ml IL-2 (Biolegend) for 48–96 h. Pan02 cells were plated in 24-well plates and stimulated with 5 μg/ml of OVA peptide for 12 h. The activated PBMCs ($2 \times 10^5$ cells/well) were added to Pan02 cells at an effector-to-

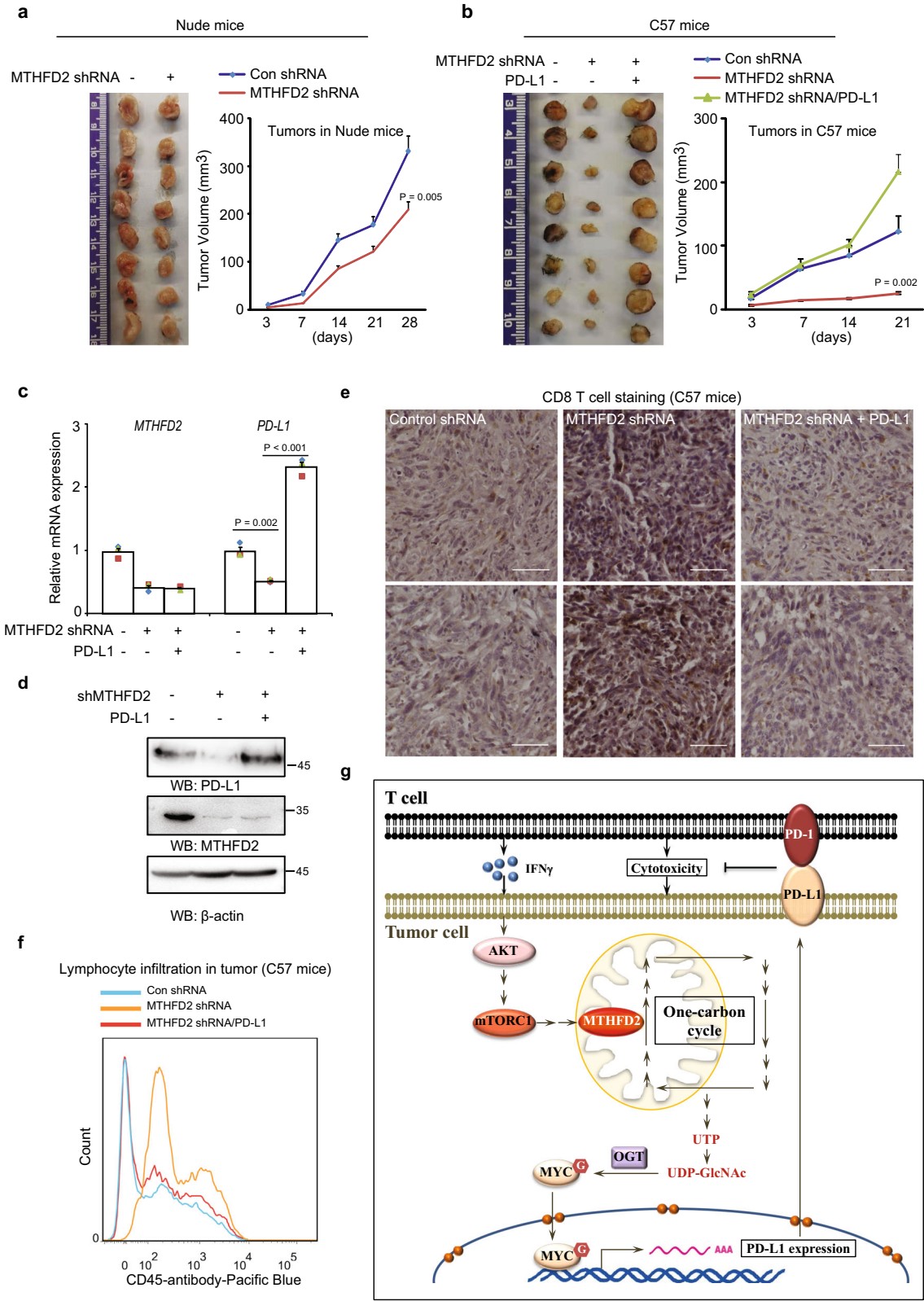

target ratio of 5 and co-cultured for 16 h. The cytotoxicity of cell killing was detected using the LDH release kit (Beyotime) described above.

**Real-Time Cell Analysis by xCELLigence Plates**. The E16 xCELLigence plates were prepared by the addition of complete media (50 µL) to each well. After equilibration at 37 °C, plates were inserted into the xCELLigence station. SW1990 cells were seeded into the wells in 100 µL culture. For T cell-killing assay, the

activated T cells were added to each well and the cell index was real-time recorded. The data were analyzed with the RTCA software 2.0 and a fulltime scale of raw data was exported.

**Immunofluorescence Analysis by FACS**. Immunofluorescence staining was performed on SW1990 cells transfected with siMTHFD2 and PD-L1 plasmid. Cells were scratched from the culture dish and washed with PBS. Each sample was

**Fig. 7 Upregulation of PD-L1 by MTHFD2 is required for tumorigenesis. a–b** A total of $2 \times 10^6$ (**a**) or $5 \times 10^6$ (**b**) PanO2 cells stably infected with lentivirus carrying indicated shRNAs or exogenous expressed PD-L1, were subcutaneously injected into athymic nude mice (**a**) or C57 mice (**b**), tumor volume was calculated every 7 days. Tumor xenografts at the 28th day in nude mice (**a**) or the 21th day in C57 mice (**b**) were shown. Data represent the means ± s.e.m ($n = 8$ mice per group); p value (Student's t test, two-sided) with control is presented. **c**, PD-L1 mRNA levels in tumor tissues in C57 mice were analyzed by real-time PCR. The values are presented as mean ± s.e.m ($n = 3$); p values (Student's t test, two-sided) with control or the indicated groups are presented. **d** The lysates of 8 pooled tumor tissues in C57 mice were subjected to immunoblotting analyses using the indicated antibodies. **e** Immunohistochemical staining was performed on tumor sections in C57 mice with anti-CD8 antibody. Representative images are shown. Scale bars, 50 μm. Histological semi-quantification was performed. **f** Cells digested from indicate tumor tissues in C57 mice were stained with anti-CD45 antibody and subjected to flow cytometric analyses (also see Supplementary Fig. 7b). Representative images (1 out of 3 experiments) are shown. **g** A schematic model showing the role of MTHFD2 in tumor immune evasion. MTHFD2 promotes PD-L1 mediated tumor immune evasion through the folate-cycle–UTP-UDP-GlcNAc metabolic pathway and the UDP-GlcNAc-O-GlcNAcylation-MYC-PD-L1 signaling pathway. Moreover, MTHFD2 could be induced by IFN-γ through AKT–mTORC1 pathway, and in turn sustain IFN-γ-upregulated PD-L1 expression.

incubated with PD-L1 antibody (abcam, clone28-8) for 30 min on ice in dark. The immunofluorescence of PD-L1 in SW1990 was analyzed by flow cytometry.

Xenografts from C57BL/6J mice were removed and minced into 1 mm³ pieces in the solution of 0.2% collagenase IV at 37 °C for 4 h with mild shake. Cells dissociated from the tissues were sieved through a 40-μm mesh and diluted in 10⁶ cells/mL in PBS. Samples were incubated with CD45 eFluor 450 (eBioscience) for 30 min in dark. Cells were acquired in a FACS Verse flow cytometer (BD), and data were analyzed using FlowJo software (Tree Star).

**Immunohistochemistry.** Formalin fixed paraffin embedded consecutive human pancreatic cancer tissue sections (3–5 μm) were deparaffinized and rehydrated. Xenografts from C57BL/6J mice were fixed with 4% PFA, dehydrated with graded ethanol, and embedded in paraffin. Tissues in 5-μm slices were deparaffinized and rehydrated.

Antigen retrieval was performed by boiling tissue slices in 10 mM sodium citrate buffer (pH 6.0) in a microwave oven for 15 min Non-specific binding sites were blocked by incubating the slides with 10% donkey serum at RT. Sections of human tumor tissues were respectively incubated with O-GlcNac, MTHFD2, PD-L1 (CST, 1:200) antibodies at 4 °C overnight, while sections of xenografts from C57BL/6J mice with rat monoclonal anti-CD8 (eBioscience, 1:200). After incubation with the primary antibodies, the sections were washed with TBST and incubated with biotinylated goat anti-mouse/rabbit/rat IgG antibody for 2 h at RT. After washing, sections were incubated with HRP-streptavidin for 1 h at RT. After DAB and counterstaining with hematoxylin, sections were mounted and imaged using a Nikon upright microscope.

Immunoreactivity was semi-quantitatively evaluated according to intensity and area: the background staining intensity was recorded as "no staining (0)", "weak to moderate staining (1)" or "strong staining (2)". The area of stained cells was recorded as <33% (1), 33–66% (2) or >66%[10] of all cells. These numbers were then multiplied resulting in a score of 0–6.

The use of human pancreas tumor specimens and the database was obtained from Shanghai East Hospital of Tongji University and approved by the Ethics Committee of the Shanghai East Hospital of Tongji University and patient consent was obtained.

**Immunoprecipitation and Western blotting.** Proteins were extracted from cultured cells or tumor tissues using cold RIPA lysis buffer (Beyotime) and protease inhibitor cocktail or phosphatase inhibitor cocktail, followed by immunoprecipitation and immunoblotting with the corresponding antibodies. Proteins from cell lysates or tumor tissues were separated by SDS-PAGE, then transferred to PVDF membrane (Bio-Rad) and probed with the indicated antibodies including MTHFD2 (CST,1:1,1000), PD-L1 (CST, 1:1,000), O-GlcNAc (CST, 1:1,000), OGT (abcam, 1:1,000), Anti-FLAG® M2 Magnetic Beads (sigma) and β-Actin−Peroxidase antibody (sigma, 1:5000), Myc (Proteintech, 1:1,000), p-STAT-1 (CST, 1:1,000) and STAT-1(CST, 1:1,000). Immunoblots were visulized using a chemi-luminescence imaging system (Tanon).

**Luciferase Reporter Assay.** Human PD-L1 promoter (from −2000 to +0 bp) was amplified from SW1990 and subcloned into the pGL4.10 vector (Promega). Luciferase activity was performed using whole-cell lysates with the Dual-Luciferase Reporter Assay System according to the manufacturer's instructions (Promega). Firefly luciferase activity was normalized to Renilla luciferase activity. Data were expressed as fold change versus control in 3 independent experiments.

**RNA sequencing.** Total RNA from SW1990 treated with siMTHFD2 was lysed with TRIzol reagents (Invitrogen) and extracted using MiniBEST Universal RNA Extraction Kit (Takara). Quality assessment, and sequencing on the RNAref +RNAseq Illumina platform were performed by BGI Genomics. Down-regulated differentially expressed genes were identified as log2 (siMTHFD2 FPKM/siControl FPKM) < −1 and 386 down-regulated differentially expressed genes were selected with Q-value (adjusted statistical significance values) less than 0.001.

**Quantitative Real-Time PCR.** Total RNA from cells or tumor tissues were extracted with TRIzol RNA isolation reagents (Invitrogen) and reversely transcripted to cDNA using the TransScript® II First-Strand cDNA Synthesis SuperMix (Transgen). Quantitative Real-time PCR was conducted with SYBR Master Mix (Yeasen). Fold changes in gene expression were calculated using $2^{-\triangle\triangle t}$ method and normalized to the expression of β-actin. The qPCR primers are shown in Supplementary Data 6.

**Recombinant protein purification.** Pet28a-PD-1 was used to transform BL21/DE3 bacteria. The cultures were grown at 37 °C to an OD 600 nm of 0.6 before inducing with 0.5 mM IPTG for 16 h at 25 °C. Cell pellets were collected and lysed by sonication. For the inclusion body purification of His-tagged PD-1, lysate sediments were denatured and mostly dissolved with a lysis buffer containing 100 mM NaH₂PO₄, 10 mM Tris, 8 M Urea, 20 mM Imidazole, pH8. The supernatants from the denatured lysates were bound to Talon metal affinity resins (Takara). Proteins were eluted with the elution buffer containing 100 mM NaH₂PO₄, 10 mM Tris-HCl, 8 M Urea, 100 mM Imidazole, pH4.5. Eluates were gradually exchanged to PBS without urea and concentrated using 3000 MWCO Ultra-15 Centrifugal Filters (Millipore).

**PD-1 binding assay.** Purified PD-1 was labeled with FITC and used for the binding assay with PD-L1. Cells were plated in glass-bottom dish. FITC labeled PD-1 was added to each well and the fluorescence images were taken with the LSM800 confocal microscope (Zeiss). PD-1 binding was evaluated with the mean fluorescence intensity normalized to the control.

**Metabolomic analyses.** SW1990 cells in dish transfected with siRNA were washed with cold PBS, scratched in extraction buffer (methanol: acetonitrile: H₂O; 2:2:1, v/v), and stored at −80 °C before analysis. The UHPLC (1290 Infinity LC, Agilent Technologies) coupled to a quadrupole time-of-flight (AB SciexTripleTOF6600) were used for metabolomics analyses in Shanghai Applied Protein Technology Co., Ltd.

A 2.1 mm × 100 mm ACQUIY UPLC BEH 1.7 μm column (waters, Ireland) was used for HILIC separation. The mobile phase (A: 25 mM ammonium acetate, 25 mM ammonium hydroxide; B: acetonitrile) used in both ESI positive and negative modes: 85% B for 1 min, then linearly reduced to 65% in 11 min, then reduced to 40% in 0.1 min and kept for 4 min, then increased to 85% in 0.1 min, with a 5 min re-equilibration.

The ESI source conditions: Gas1 60, Gas2 60, CUR 30, source temperature: 600 °C, Ion Spray Voltage Floating (ISVF) ± 5500 V. In MS only acquisition, m/z range 60–1000 Da, accumulation time for TOF MS scan 0.20 s/spectra. In auto MS/MS acquisition, m/z range 25–1000 Da, accumulation time 0.05 s/spectra. The ion scan is acquired using information dependent acquisition (IDA) with high sensitivity mode and parameters: the collision energy (CE) 35 V with ±15 eV; declustering potential (DP) 60 V (+) and −60 V (−); exclude isotopes within 4 Da, candidate ions to monitor per cycle: 10.

The raw data (wiff.scan files) were converted to MzXML files using ProteoWizard MSConvert and then imported into XCMS software. The parameters for peak picking: centWave m/z 25 ppm, peak width c (10, 60), prefilter c (10, 100). The parameters for peak grouping: bw 5, mzwid 0.025, minfrac 0.5. For annotation of isotopes and adducts, CAMERA (Collection of Algorithms of MEtabolitepRofile Annotation) was used. For ion features, only the variables having more than 50% of the nonzero measurement values in at least one group were kept. Metabolite identification was performed by comparing accuracy m/z value (<25 ppm); MS/MS spectra with an in-house database established with available authentic standards.

**Statistics.** Data were presented as mean ± s.e.m. Statistical significance was set at P < 0.05. Significance between two groups was determined using the Student's t test (unpaired two-tailed, unequal variance). The Kaplan–Meier (KM) survival analysis was performed by comparing the survival data from the databases include GEO,

EGA, and TCGA as described[42]. The statistical analysis was performed in GraphPad Prism 7. All experiments were repeated at least three times unless otherwise indicated. *N* numbers are indicated in the figure legends.

**Reporting summary**. Further information on research design is available in the Nature Research Reporting Summary linked to this article.

## Data availability

Supplementary Figs. 1c–i, 2b, c, e, 3a–i, 4a–f, 5a–c, 6a–d have been provided as Source Data File (an excel file contains 77 sheets). The source information for other figures and supplementary figures have also been provided as below: The source information for Figs. 1a, b and Supplementary Fig. 1a have been provided in Supplementary Data 1 (an excel file contains 4 sheets) and Supplementary Data 2 (an excel file), and the related gDNA-seq data have been deposited in the NCBI database with accession code PRJNA670077. The source information for Fig. 2a, b have been provided in Supplementary Data 3 (an excel file) and Supplementary Data 4 (an excel file), and the related RNA-seq data have been deposited in the NCBI database with accession code PRJNA670286. The source information for Figs. 4c, 5a have been provided in Supplementary Data 5 (an excel file contains 3 sheets), and the related mass spectrometry metabolomic data have been deposited to the ProteomeXchange Consortium via the iProX partner repository with the dataset identifier PXD024228. A complete list of all primers used has been supplied as Supplementary Data 6. The data in Fig. 1f were obtained and analyzed in the KM plot website [http://kmplot.com/analysis/index.php?p=service&cancer=pancancer_rnaseq]. The data in Fig. 2b and Supplementary Fig. 2a and Supplementary Data 4 were obtained and analyzed in cbioportal website [http://www.cbioportal.org; more specific hyperlink has been provided in Supplementary Data 4]. Any other information supporting the findings of this study is available from the corresponding author on reasonable request. Source data are provided with this paper.

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

## Acknowledgements

This work was supported by National Nature Science Foundation of China 81874055 (T.W.), 82073051 (T.W.), 81872239 (Q.Y.), 31800710 (Y.F.), 81602042 (T.W.), 31701220 (K.Z.), 81672710 (Q.Y.); Academician expert workstation grant 19R6002099223 (K.Z.) in Shanghai Yuansong Bio-technology Limited Company; Tianjin Research Innovation Project for Postgraduate Students (H.Y.).

## Author contributions

This study was conceived by T.W. and Q.Y.; T.W., M.S., and H.Y. designed the study; M.S., H.Y., R.Y., T.C., Y.F., Y.L., X.F., J.Z., H.L., X.C., J.G., and J.X. performed the experiments; T.C., Q.Z., X.L., and K.Z. provided materials; T.W., Q.Y., Y.F., and K.Z. provided funding; X.L. and K.Z. provided conceptual advice; T.W. wrote the paper with comments from all authors.

## Competing interests

The authors declare no competing interests.
