## [Peer Review File · Nature Communications]

Reviewers' comments:

Reviewer #1 (Remarks to the Author); expert on O-GlcNAcylation:

This manuscript is a well-performed and expansive study linking cancer cell metabolism to immune evasion through O-GlcNAc modification. The experiments are well performed and quite convincing to me at each stage. In particular the use of the c-myc point mutant that blocks O-GlcNAc modification is quite convincing from a mechanistic standpoint. I believe that this paper will make a nice addition to Nat Commun, but I do have 3 comments.

Comment #1 - In Figure 4a the authors observe some increase in PDL1 upon overexpression of the catalytically inactive version of MTHFD2. What are the potential explanations for this?

Comment #2 - The data Figure 5b suggests that under MTHFD2 knockdown UDP becomes the limiting metabolite for the synthesis of UDP-GlcNAc and subsequent O-GlcNAc modification. The authors should consider testing this directly by seeing if increasing glucose (or glutamine) concentrations in the media can still drive an increase in O-GlcNAc. I would expect that siControl cells would show an increase while siMTHFD2 would show a smaller increase or none at all over their respective baseline levels.

Comment #3 - It is notable that MTHFD2 loss in T cells makes them unable to kill MTHFD2-KO cancer cells with greater efficiency compared to wild-type cancer cells, while the basal level of cytotoxicity for the wild-type cells is similar at ~20%. For example, compare Figure 1c with Figure S7d. Do the authors have any explanation for why MTHFD2 expression in T cells would not appear to be super important for killing wild-type cancer cells but then show no improvement in the KOs?

Reviewer #2 (Remarks to the Author); expert on CRISPR screening:

In this manuscript, the authors performed a CRISPR screen to identify metabolic genes that confer immune resistance to cancer cells. Among the hits, they focused their analysis on MTHFD2 and found that MTHFD2 increases PD-L1 expression and the global O-GlcNAcylation level through upregulation of UDP-GlcNAc, which leads to stabilization of cMYC. Down-regulation of MTHFD2 by RNAi resulted in suppression of tumor growth in immune-competent mice and the concomitant expression of PD-L1 cancelled the effect of MTHFD2 downregulation, supporting the role of MTHFD2 in PD-L1 upregulation.

This is an interesting finding; however, during the assessment of the CRISPR screen data, I found a potentially critical data processing error and therefore I will focus my comments only on the CRISPR screening part and will not assess the follow-up part of the study.

According to the Methods, the authors seem to have performed their screen appropriately. Looking at the normalized counts, the library complexity looks reasonably maintained throughout the experiments. There are some missing details in Methods: how many mutagenized cells were used for T-cell-killing assay and how much cell death can be induced with this effector-to-target ration of 2. These information would be useful for researchers to reproduce this screen.

The CRISPR data analysis in this study is quite unusual. Firstly, I wonder what was the reason why the authors selected sgRNAs based on the criteria the authors set. It would be reasonable to remove sgRNAs that are particularly low-representation (eg, less than 10 in the control samples) but the way the authors selected seems unusual. I would suggest that the authors re-analyze their screen data using widely used statistical packages, such as MAGeCK and investigate how many genes can be re-identified

Secondly, the data files the authors provided via dropbox most likely have a fundamental error. File

'excel1' contains 3 spreadsheets, 1. Normalized sgRNA counts, 2. Statistical calculation, and 3. Selected 713 genes. The error is found in Spreadsheet 2. For example, in Spreadsheet 1 'normalized read counts', the values of 3 selected sgRNAs for MTHFD2 (#2,8 and 9) are:

```
sgRNA Gene control1 control2 Tcell1 Tcell2
sgMTHFD2_2 MTHFD2 3683.051991 4144.10202 1692.278952 2675.204229
sgMTHFD2_9 MTHFD2 2380.078826 1848.247569 2118.164036 991.0225865
sgMTHFD2_8 MTHFD2 1143.45424 779.5736781 415.6474618 423.1332392
```

However, in Spreadsheet 2 'statistical calculation', the values of these sgRNAs are:

```
sgRNA Gene con1 con2 Tcell1 Tcell2
sgMTHFD2_2 MTHFD2 3683.05199 4144.10202 632.685053 1471.6871
sgMTHFD2_9 MTHFD2 2380.07883 1848.24757 1361.60375 1261.97633
sgMTHFD2_8 MTHFD2 1143.45424 779.573678 881.459272 1009.58106
```

Between the two spreadsheets, the values for the controls are the same, but those for the Tcell are inconsistent. This inconsistency can be seen almost the entire sgRNAs listed in Spreadsheet2 (2589 inconsistency out of the 2590 sgRNA selected). All downstream calculations are based on the read counts of the selected sgRNAs listed in Spreadsheet 2 and therefore are most likely all wrong. I suppose the selection of sgRNAs is also wrongly done.

Even if correct read counts are analyzed, the downstream analyses the authors made are problematic: in particular, taking an average of read counts of all sgRNAs targeting the same gene and making this average count as a gene-level count is not appropriate. Gene-level fold change should not be calculated using values obtained in this way. To calculate gene-level fold changes, the authors should first calculate a sgRNA-level fold change for each sgRNA and then take a mean among the sgRNA-level fold change values of the same gene.

The gene selection process summarized in File 'excel2' is based on the outcome of File 'excel1'. Therefore, the Venn diagrams shown in Figure 1a would be completely different when the correct data are analyzed. In my re-analysis for MTHFD2 with the correct counts using the authors' data analysis workflow, this gene still pass the z-score filter but does not pass the FC filter.

Additional comment I have is regarding Data Availability statement. All data must be available without contacting or making a request to the authors. For CRISPR data, the authors must include sgRNA read counts as supplementary data. Gene expression data must be deposited in publicly available repository (eg GEO), which I believe every journal asks and is prerequisite for submission.

Reviewer #3 (Remarks to the Author); expert on immunology and metabolism:

In this manuscript, the authors have explored how cancer metabolism contributed to tumor immune evasion. The authors found MTHFD2, a folate cycle enzyme, promoted PD-L1 expression via MYC protein stabilization, and MTHFD2 is induced by IFN γ through AKT/mTOR signaling and involved in IFN γ -induced PD-L1 expression. The work is potentially interesting. However, it suffers from many major weaknesses.

1. The major conclusion that "MTHFD2 promotes tumor immune evasion via upregulating PD-L1 expression" is insufficiently convincing. The authors observed PD-L1 overexpression could reduce the cytotoxicity of T cells in MTHFD2 KO cells (Fig.2I). PD-L1 overexpressing and PD-L1 knockout MTHFD2 WT cells are essential controls and must be included. These controls should be included in many other experiments when appropriate.
2. To link MTHFD2 and tumor PD-L1 to tumor immunity, PD-L1 knock out cells should be utilized. In PD-L1 KO cells, can manipulation of MTHFD2 affect T cell immunity in vitro and in vivo?
3. As an important metabolic enzyme, MTHFD2 may have its intrinsic role in cell biology. The metabolic pattern of MTHFD2 knock out/ knock down cells may have been evolved in the course of cell line establishment. Using stable cell lines to study the metabolic function of MTHFD2 is questionable. At least, this approach should be complemented and validated in the Tet-on system. Accordingly, the in vivo and in vitro experiments are needed.

4. The effect of IFN γ on MTHFD2 is not clear. The authors may need to use JAK1/ STAT1 KO cells to verify that the regulation of MTHFD2 is independent of canonical JAK/STAT pathway.
5. The authors proposed that AKT/mTOR inhibitors blocked IFN γ signaling to MTHFD2, thereby regulating PD-L1. However, it is possible that AKT/mTOR inhibitor regulates PD-L1 in an MTHFD2-independent manner. If MTHFD2 KO cells were treated with AKT/mTOR inhibitor, will STAT1 phosphorylation/ IRF1 expression/ PD-L1 expression be affected upon IFN γ treatment?
6. In general, PD-L1 transcription is directed by STAT1/IRF1, when treated with AKT/mTOR inhibitor, the IFN γ signaling through STAT1/IRF1 may also be affected. To uncouple the effect of STAT1/IRF1 on PD-L1, the authors may need to employ STAT1/IRF1 Knock out cells. If STAT1/IRF1 Knock out cells were treated with IFN γ /siMTHFD2, will the PD-L1 expression be affected?
7. The human CTL-killing assay is problematic. When human cancer cells were cultured with human activated CD8+ T cells, the cytotoxicity may not be antigen specific and may not be affected by tumor cell PD-L1 expression. Again, it suggests an alternative role of MTHFD2 independent of PD-L1.
8. In Fig.3a, the RNA levels of PD-L1 is induced by IFN γ around 2-fold (Fig.3a), but the protein levels are dramatically induced by IFN γ (Fig.3b), this raises the concern that IFN γ may not mainly regulate PD-L1 expression at transcription level, but at translation or protein stability.
9. In Fig.3a, the PD-L1 regulation by IFN γ is not consistent. In MCF-7 and HepG2 cells, the fold change of PD-L1 induced by IFN γ is not affected by siMTHFD2, suggesting that MTHFD2 is not involved in the regulation of PD-L1 by IFN γ . Furthermore, in Fig.3d, the effect of LY294002 on PD-L1 is weak.
10. How is MTHFD2 regulated by AKT/mTOR pathway?
11. The authors have completely ignored the importance of PD-L1 (B7-H1) in antigen presenting cells, particularly macrophages and dendritic cells.

Reviewer #4 (Remarks to the Author); expert on one-carbon metabolism and immunology:

The study 'Metabolic control of immune evasion via tumour MTHFD2' identifies the folate cycle enzyme MTHFD2 as a factor that can dampen effector T cell cytotoxicity and therefore fosters tumor growth. The authors show that MTHFD2 supports PD1L expression by a mechanism that involves O-GlcNAcylation of cMYC.

This is an interesting paper and most of the experiments are well performed. I especially would like to highlight that the key findings of the current study are supported by various 'rescue experiments' (mutants, treatment with UDP/UTP, inhibitors, reexpression) basically on every level. In addition, findings by the authors are supported by correlations found by the analyses of patient material.

Specific points:

- In figure 3d – 3e, the authors should add IFN γ with Rapamycin and/or LY294002 on the MTHFD2 KD and KO cells to show that the inhibitors do not have an effect of PD-L1 and that this effect is dependent on the presence of MTHFD2.
- In figure 5a, siControl5 and siControl6 look completely different than the rest of the controls, why is that?
- The study would benefit from more physiological data that shows the the significance of O-GlcNAcylation. How does expression of Myc(tag)-OGT expression (in tumors) affect T cell killing?...and/or even tumor growth.
- In the text related to figure 5, the authors conclude that the MTHFD2-enhanced PD-L1 expression is mediated by O-GlcNAcylation. However, to confirm that, they should inhibit O-GlcNAcylation (blocking OGT) and then see if the expression of PD-L1 is affected.
- In figure 7g, they use adenovirus treatment as a potential treatment strategy. I found this a bit stretched and the authors do not really need this data.

Reviewers' comments:

Reviewer #1 (Remarks to the Author); expert on O-GlcNAcylation:

This manuscript is a well-performed and expansive study linking cancer cell metabolism to immune evasion through O-GlcNAc modification. The experiments are well performed and quite convincing to me at each stage. In particular the use of the c-myc point mutant that blocks O-GlcNAc modification is quite convincing from a mechanistic standpoint. I believe that this paper will make a nice addition to Nat Commun, but I do have 3 comments.

Answer: We greatly appreciate the reviewer's acknowledgement of the significance of our report and the insightful comments that will certainly strengthen our manuscript.

Comment #1 - In Figure 4a the authors observe some increase in PDL1 upon overexpression of the catalytically inactive version of MTHFD2. What are the potential explanations for this?

Answer: That's a very good question. Asp168 of MTHFD2 contributes to a Mg²⁺ binding site, and the D168E mutation highly inhibits the dehydrogenase activity of MTHFD2 (J Biol Chem. 2005 Oct 7;280(40):34316-23). However, as the reviewer pointed out, we definitely found that the overexpressed MTHFD2-D168E mutant also slightly but significantly increased PD-L1 expression (Fig. 4a). We speculate that was most likely due to the highly but not fully inhibited catalytic activity in MTHFD2-D168E mutant. However, based on this data, we can't fully rule out the possibility that a catalytic activity independent role of MTHFD2 may be involved in this process, thus, we make a conclusion in the manuscript that "its catalytic activity is an important contributor to PD-L1 expression".

Comment #2 - The data Figure 5b suggests that under MTHFD2 knockdown UDP becomes the limiting metabolite for the synthesis of UDP-GlcNAc and subsequent O-GlcNAc modification. The authors should consider testing this directly by seeing if increasing glucose (or glutamine) concentrations in the media can still drive an increase in O-GlcNAc. I would expect that siControl cells would show an increase while siMTHFD2 would show a smaller increase or none at all over their respective baseline

levels.

Answer: This question is well taken. We did experiments based on reviewer's suggestion to analyze the total O-GlcNAc modification (Fig. S5a) in the revised manuscript. To enhance the effect of glucose, we performed that in a limited glucose condition (2.5 mM) which highly decreased the basal O-GlcNAc. After high glucose (25 mM) stimulation, we still detected a notable increase in siMTHFD2 cells but obviously much lower than that in siControl cells, which suggested both glucose and nucleotide (promoted by MTHFD2) could be the limiting metabolite for O-GlcNAc modification.

Comment #3 - It is notable that MTHFD2 loss in T cells makes them unable to kill MTHFD2-KO cancer cells with greater efficiency compared to wild-type cancer cells, while the basal level of cytotoxicity for the wild-type cells is similar at ~20%. For example, compare Figure 1c with Figure S7d. Do the authors have any explanation for why MTHFD2 expression in T cells would not appear to be super important for killing wild-type cancer cells but then show no improvement in the KOs?

Answer: Thanks for a careful review and pointing out this issue. We fully agree with the reviewer that, in theory, the siMTHFD2 CD8⁺ T cells should display a weaker cytotoxicity. However, each time of cytotoxicity assay, the fresh PBMCs were isolated from blood of different healthy donors. And the CD8⁺ T cells were further isolated from PBMCs and activated by anti-CD3, anti-CD28 and IL-2 *in vitro* to acquire killing ability. Although we try the best to perform this process exactly the same, the CD8⁺ T-cells were from different donors and activated by different batches of cytokines (anti-CD3, anti-CD28 and IL-2). That's highly possible to display a different basal level of cytotoxicity, such as from 5.48% (Fig. 7f in the unrevised manuscript) to 29.2% (Fig. 1d). Thus, we think to compare the cytotoxicity level induced by CD8⁺ T cells from the same donors and activated at the same batch should be more suitable.

During the process for establishing a MTHFD2-targeted strategy for cancer treatment, we found that the MTHFD2 is also involved in T cell function. Therefore, we have generated a practical strategy, which affects tumor MTHFD2 but not T cell MTHFD2.

We were very excited that this strategy exhibited a promising effect in human tumors under humanized immunity *in vivo*. However, another reviewer think this part was out of the general focus this study and suggested us to delete this part. Although a little pity, we have removed this part of data including the figure S7d (discussed here) in the revised manuscript.

Reviewer #2 (Remarks to the Author); expert on CRISPR screening:

In this manuscript, the authors performed a CRISPR screen to identify metabolic genes that confer immune resistance to cancer cells. Among the hits, they focused their analysis on MTHFD2 and found that MTHFD2 increases PD-L1 expression and the global O-GlcNAcylation level through upregulation of UDP-GlcNAc, which leads to stabilization of cMYC. Down-regulation of MTHFD2 by RNAi resulted in suppression of tumor growth in immune-competent mice and the concomitant expression of PD-L1 cancelled the effect of MTHFD2 downregulation, supporting the role of MTHFD2 in PD-L1 upregulation.

This is an interesting finding; however, during the assessment of the CRISPR screen data, I found a potentially critical data processing error and therefore I will focus my comments only on the CRISPR screening part and will not assess the follow-up part of the study.

According to the Methods, the authors seem to have performed their screen appropriately. Looking at the normalized counts, the library complexity looks reasonably maintained throughout the experiments. There are some missing details in Methods: how many mutagenized cells were used for T-cell-killing assay and how much cell death can be induced with this effector-to-target ration of 2. These information would be useful for researchers to reproduce this screen.

The CRISPR data analysis in this study is quite unusual. Firstly, I wonder what was the

reason why the authors selected sgRNAs based on the criteria the authors set. It would be reasonable to remove sgRNAs that are particularly low-representation (eg, less than 10 in the control samples) but the way the authors selected seems unusual. I would suggest that the authors re-analyze their screen data using widely used statistical packages, such as MAGeCK and investigate how many genes can be re-identified. Secondly, the data files the authors provided via dropbox most likely have a fundamental error. File 'excel1' contains 3 spreadsheets, 1. Normalized sgRNA counts, 2. Statistical calculation, and 3. Selected 713 genes. The error is found in Spreadsheet 2. For example, in Spreadsheet 1 'normalized read counts', the values of 3 selected sgRNAs for MTHFD2 (#2,8 and 9) are:

sgRNA	Gene	control1	control2	Tcell1	Tcell2
sgMTHFD2_2	MTHFD2	3683.051991	4144.10202	1692.278952	2675.204229
sgMTHFD2_9	MTHFD2	2380.078826	1848.247569	2118.164036	991.0225865
sgMTHFD2_8	MTHFD2	1143.45424	779.5736781	415.6474618	423.1332392

However, in Spreadsheet 2 'statistical calculation', the values of these sgRNAs are:

sgRNA	Gene	con1	con2	Tcell1	Tcell2
sgMTHFD2_2	MTHFD2	3683.05199	4144.10202	632.685053	1471.6871
sgMTHFD2_9	MTHFD2	2380.07883	1848.24757	1361.60375	1261.97633
sgMTHFD2_8	MTHFD2	1143.45424	779.573678	881.459272	1009.58106

Between the two spreadsheets, the values for the controls are the same, but those for the Tcell are inconsistent. This inconsistency can be seen almost the entire sgRNAs listed in Spreadsheet2 (2589 inconsistency out of the 2590 sgRNA selected). All downstream calculations are based on the read counts of the selected sgRNAs listed in Spreadsheet 2 and therefore are most likely all wrong. I suppose the selection of sgRNAs is also wrongly done.

Even if correct read counts are analyzed, the downstream analyses the authors made are problematic: in particular, taking an average of read counts of all sgRNAs targeting the same gene and making this average count as a gene-level count is not appropriate.

Gene-level fold change should not be calculated using values obtained in this way. To calculate gene-level fold changes, the authors should first calculate a sgRNA-level fold change for each sgRNA and then take a mean among the sgRNA-level fold change values of the same gene.

The gene selection process summarized in File 'excel2' is based on the outcome of File 'excel1'. Therefore, the Venn diagrams shown in Figure 1a would be completely different when the correct data are analyzed. In my re-analysis for MTHFD2 with the correct counts using the authors' data analysis workflow, this gene still pass the z-score filter but does not pass the FC filter.

Additional comment I have is regarding Data Availability statement. All data must be available without contacting or making a request to the authors. For CRISPR data, the authors must include sgRNA read counts as supplementary data. Gene expression data must be deposited in publicly available repository (eg GEO), which I believe every journal asks and is prerequisite for submission.

Answer: We express our especial and sincere gratitude to the reviewer, an excellent expert in bioinformatics, who pointed out a critical mistake in our manuscript and protect us from an potential disaster. During processing the CRISPR screen data (previous Fig. 1a), the sgRNA name and the read counts have been mismatched for some ridiculous reason. There is no any excuse for a mistake like this. I feel very sorry for this irresponsible behavior. I sincerely apologize for that.

Therefore, we reanalyzed this data with the help of an expert in bioinformatics using MAGeCK software as suggested, and revised the Fig. 1 and S1. It's noted that, if the counts in same repeats showing more than 5 fold-change, we think the value cannot be trusted and should be rule out. In the revised Fig. 1, our focused gene MTHFD2 in Z-score analysis has also been identified. Most importantly, the MTHFD2 show a clear and strong effect in all the following functional studies. However, the other 2 genes MTHFD1 and MTHFD1L cannot be screened out as the previous wrong analysis. This is also in consistent with our previous functional verification in unrevised Fig. 1c that

these two genes show much less functional effect than MTHFD2. This was also the reason that all our following studies only focused on MTHFD2. Here we revised the CRISPR data, and removed few information related to MTHFD1/1L in Fig. 1 and S1.

A file contains the data during each step of the analysis were attached as supplementary information (Excel S1-Metabolic CRISPR library screen) in the revised manuscript:

Sheet 1: The unprocessed sgRNA counts

Sheet 2: The sgRNA counts in same repeats showing more than 5 fold-change were removed

Sheet 3: The analyzed information of 3261 genes by MAGeCK (Z-scores)

Sheet 4: The 72 genes with highly decreased sgRNAs after T cell treatment (Z score < -3, P < 0.1)

The original raw data for CRISPR screen (fastq) has been deposited in NCBI database: SUB8324077

(<https://dataview.ncbi.nlm.nih.gov/object/PRJNA670077?reviewer=s2fjbq2vjfl7fegtbvgkgqh118>)

The original raw data for the RNAseq has been deposited in NCBI database: SUB8336473

(<https://dataview.ncbi.nlm.nih.gov/object/PRJNA670286?reviewer=emqb9scu19r35dtclk66dl958s>)

Reviewer #3 (Remarks to the Author); expert on immunology and metabolism:

In this manuscript, the authors have explored how cancer metabolism contributed to tumor immune evasion. The authors found MTHFD2, a folate cycle enzyme, promoted PD-L1 expression via MYC protein stabilization, and MTHFD2 is induced by IFN γ through AKT/mTOR signaling and involved in IFN γ -induced PD-L1 expression. The work is potentially interesting. However, it suffers from many major weaknesses.

Answer: We greatly appreciate the reviewer's acknowledgement of the significance of our report and the insightful comments that will certainly strengthen our manuscript.

1. The major conclusion that "MTHFD2 promotes tumor immune evasion via upregulating PD-L1 expression" is insufficiently convincing. The authors observed PD-

L1 overexpression could reduce the cytotoxicity of T cells in MTHFD2 KO cells (Fig.2l). PD-L1 overexpressing and PD-L1 knockout MTHFD2 WT cells are essential controls and must be included. These controls should be included in many other experiments when appropriate.

Answer: Thanks for the reviewer's suggestion that "PD-L1 overexpressing and PD-L1 knockout MTHFD2 WT cells are essential controls and must be included ". We have moved the previous Fig.2l into Fig. S2 and performed additional experiments based on reviewer's suggestion in the revised manuscript (Fig. 2l, 2m, 2n and S2e). T cell cytotoxicity assay based on LDH release (Fig. 2l) showed aggravated cytolysis in MTHFD2 KD cancer cells, which was significantly reversed by the exogenously expressed PD-L1. In addition, T cell-induced cell death was aggravated by either MTHFD2 or PD-L1 depletion but only with a very mild additional effect when both were simultaneously depleted (Fig. 2m and S2e). Accordingly, overexpressed MTHFD2 show a compromised effect in cancer cells with PD-L1 depletion (Fig. 2n). These data suggested that MTHFD2 protects tumour cells from effector T cell cytotoxicity mainly through PD-L1.

2. To link MTHFD2 and tumor PD-L1 to tumor immunity, PD-L1 knock out cells should be utilized. In PD-L1 KO cells, can manipulation of MTHFD2 affect T cell immunity in vitro and in vivo?

Answer: Thanks for the good suggestion. We have performed additional experiments based on reviewer's suggestion with PD-L1 KD in the revised manuscript (Fig. 2m, 2n and S2e). T cell-induced cell death was aggravated by either MTHFD2 or PD-L1 depletion but only with a very mild additional effect when both were simultaneously depleted (Fig. 2m and S2e). Accordingly, overexpressed MTHFD2 show a compromised effect in cancer cells with PD-L1 depletion (Fig. 2n). These data suggested that MTHFD2 protects tumour cells from effector T cell cytotoxicity mainly through PD-L1.

3. As an important metabolic enzyme, MTHFD2 may have its intrinsic role in cell biology. The metabolic pattern of MTHFD2 knock out/ knock down cells may have been evolved in the course of cell line establishment. Using stable cell lines to study the metabolic function of MTHFD2 is questionable. At least, this approach should be

complemented and validated in the Tet-on system. Accordingly, the in vivo and in vitro experiments are needed.

Answer: Thanks for this suggestion. MTHFD2 is important for efficient folate-cycle especially in multiple human tumours. In basal cell biology, MTHFD2 has a functional substitute MTHFD2L with 50-fold lower activity, and MTHFD2 is absent in basal but inducible in most cells. Tet-on system is a method of inducible gene expression where transcription is turned on in the presence of the antibiotic tetracycline or its derivatives doxycycline. However, tetracycline/doxycycline could interfere with folic acid (vitamin B9) (Gastroenterology. 1966 Sep;51(3):317-32.), a metabolite critical for folate-cycle metabolism. Thus, folic acid should not be taken at the same time with the tetracycline when supplementing folic acid. There is possibility that tetracycline may affect the functional readout of MTHFD2. Therefore, Tet-on system is not a good choice for studying the role of MTHFD2. Instead, transient knock down by siRNA has been performed at each step during mechanism exploring.

4. The effect of IFN γ on MTHFD2 is not clear. The authors may need to use JAK1/STAT1 KO cells to verify that the regulation of MTHFD2 is independent of canonical JAK/STAT pathway.

Answer: Thanks for this point. To emphasize the STAT1-independent regulation of MTHFD2, we have performed additional experiments with STAT1 inhibitor Fludara in the revised manuscript (Fig. S3d). The same as siSTAT1 (Fig. S3c), the STAT1 inhibitor Fludara fully eliminated both basal and IFN γ -stimulated p-STAT1 (Fig. S3d). However, in Fig. S3c and Fig. S3d, both STAT1 siRNA and inhibitor displayed no effect on MTHFD2 level no matter basal or IFN γ -treated conditions, suggesting MTHFD2 is not the downstream of STAT1 no matter basal or IFN γ -treated conditions.

5. The authors proposed that AKT/mTOR inhibitors blocked IFN γ signaling to MTHFD2, thereby regulating PD-L1. However, it is possible that AKT/mTOR inhibitor regulates PD-L1 in an MTHFD2-independent manner. If MTHFD2 KO cells were treated with AKT/mTOR inhibitor, will STAT1 phosphorylation/ IRF1 expression/ PD-L1 expression be affected upon IFN γ treatment?

Answer: This question is well taken. We have performed the experiment based on the reviewer's suggestion in the revised manuscript (Fig. S3f). Rapamycin obviously

inhibited the IFN- γ -induced MTHFD2 and PD-L1; while PD-L1 was also slightly suppressed by rapamycin in MTHFD2 KD cells, suggesting mTORC1 stimulates PD-L1 only partially through MTHFD2. However, depleting MTHFD2 has no effect on p-STAT1 no matter basal or IFN γ -treated conditions, suggesting STAT1 is not the downstream of MTHFD2 no matter basal or IFN γ -treated conditions.

According to our data related to IFN γ (Fig. 3), we emphasize several points about the relationship among IFN γ , AKT/mTOR, MTHFD2 and PD-L1 with the diagram below.

① The basal tumor MTHFD2 upregulating PD-L1 is the key finding in this study (red in the diagram). ② IFN γ upregulating MTHFD2 was an unexpected finding during we analyzing the role of MTHFD2 in IFN γ -induced PD-L1 expression. ③ MTHFD2 KD suppressed both basal and IFN- γ -elevated PD-L1, indicating high basal level of tumor MTHFD2 plus IFN- γ -elevated MTHFD2 both contribute to PD-L1 expression during IFN- γ stimulation. ④ IFN- γ induced notable PD-L1 expression within 4h, but MTHFD2 required 12-24h. This suggests that tumor basal MTHFD2 but not IFN- γ -elevated MTHFD2 is involved in the initial process of IFN- γ -induced PD-L1 expression and the IFN γ -elevated MTHFD2 is just one of the contributors for PD-L1 expression. ⑤ mTORC1 pathway promotes PD-L1 expression highly but not fully dependent on MTHFD2. ⑥ STAT1, the canonical downstream of IFN- γ and known to promotes PD-L1 expression, was neither the major upstream nor major downstream of MTHFD2. The MTHFD2-independent ways (purple in the diagram) for IFN γ regulating PD-L1 are also important but out of the focus in this study (MTHFD2).

6. In general, PD-L1 transcription is directed by STAT1/IRF1, when treated with AKT/mTOR inhibitor, the IFN γ signaling through STAT1/IRF1 may also be affected.

To uncouple the effect of STAT1/IRF1 on PD-L1, the authors may need to employ STAT1/IRF1 Knock out cells. If STAT1/IRF1 Knock out cells were treated with IFN γ /siMTHFD2, will the PD-L1 expression be affected?

Answer: This question is well taken. We fully agree with the reviewer that PD-L1 transcription is directed by STAT1. In revised Fig. S3c, almost fully depleting STAT1 highly decreased both basal and IFN γ -induced PD-L1, suggesting STAT1 is not only important for IFN γ signaling but also important for IFN γ -independent PD-L1 basal expression. There is no any doubt about the important role of STAT1 in PD-L1 transcription. However, in the siSTAT1 cells, IFN γ still stimulated relative level of PD-L1, which was lower than that in siControl cells. This suggested there must be other mechanisms also contribute to IFN γ -induced PD-L1 (also see answer for comment 8). In revised Fig. S3f, depleting MTHFD2 has no obvious effect on p-STAT1 no matter basal or IFN γ -treated conditions, suggesting STAT1 is not the major downstream of MTHFD2 no matter basal or IFN γ -treated conditions. In revised Fig. S3c and Fig. S3d, both STAT1 siRNA and STAT1 inhibitor displayed no effect on MTHFD2 level no matter basal or IFN γ -treated conditions, suggesting MTHFD2 is not the downstream of STAT1.

In addition, we have performed the experiments in STAT1 KD cells or cells treated with STAT1 inhibitor Fludara based on the reviewer's suggestion in the revised manuscript (Fig. S3g and S3h). IFN- γ still displayed certain effect on PD-L1 induction in STAT1 KD cells (iFig. S3c) as the same in Fig. S3c and or cells treated with STAT1 inhibitor Fludara (Fig. S3h) but very weak or undetectable effect in cells with both STAT1 and MTHFD2 inhibition (Fig. S3g and S3h), which also suggested STAT1 and MTHFD2 could independently contribute to PD-L1 expression. The same as the comment 5, The MTHFD2 independent or weakly related mechanisms for IFN γ regulating PD-L1, such as STAT1, are also very important but out of the focus (MTHFD2) in this study.

7. The human CTL-killing assay is problematic. When human cancer cells were cultured with human activated CD8+ T cells, the cytotoxicity may not be antigen specific and may not be affected by tumor cell PD-L1 expression. Again, it suggests an alternative role of MTHFD2 independent of PD-L1.

Answer: This is a very important question. Plenty of studies indicated that the PD-L1/PD-1 signaling is involved in various role of T cells, such as "Engagement of PD-1

by PD-L1 alters the activity of T cells in many ways, inhibiting T cell proliferation, survival, cytokine production, and other effector functions (Butte et al., 2007; Chang et al., 1999; Curiel et al., 2003; Dong et al., 1999; Freeman et al., 2000; Keir et al., 2006; Latchman et al., 2004)." (Immunity. 2018 Mar 20;48(3):434-452). Here, we performed the human CTL-killing assay in a well established protocol used by plenty of studies, which activated CD8⁺ T cells *in vitro* mainly by anti-CD3, anti-CD28 and IL-2. The activated CD8⁺ T cells will display a broad toxicity to cancer cells with MHC1 expression. The expression of MHC1 in all the cancer cell line used in this study were proved by previous report (Nature. 2020 May;581(7806):100-105; World J Gastroenterol 2002;8(4):654-657). The expression of PD-1 in CD8⁺ T cells activated in this way has been proved by plenty of reports such as "As expected, exposure of CD8⁺ T cells to anti-CD3/anti-CD28 versus IL-2 alone *in vitro* revealed the generation of cells with markedly different phenotypes; CD3 stimulation led to high levels of CD25 and PD-1 expression, whereas IL-2 stimulation alone did not, similar to what was observed with *in vivo* activated T cells" (Blood. 2012 Mar 29; 119(13): 3073–3083). In addition, we have also detected a quite high level of PD-1 especially in activated CD8⁺ T cells in our study (Fig. S2b). Other reports using the same CTL-killing assay protocol also suggested PD-L1/PD-1 signaling is involved in this process, such as "we provide evidence that the combination of a novel PD-1–blocking antibody (cPD-1) with (E1)-3s or the CEACAM5-targeting (14)-3s could potentiate the antitumor activity of redirected T cells against target human cancer cells grown *in vitro* as monolayer cultures or three-dimensional (3D) multicellular tumor spheroids (MCTS; ref. 29)" (Cancer Res. 2017 Oct 1;77(19):5384-5394).

Of course, only based on our data, we can't fully rule out the possibility of an alternative role of MTHFD2 independent of PD-L1. We can only state that "tumor MTHFD2 facilitates T immune evasion mainly through upregulating PD-L1".

8. In Fig.3, the RNA levels of PD-L1 is induced by IFN γ around 2-fold (Fig.3a), but the protein levels are dramatically induced by IFN γ (Fig.3b), this raises the concern that IFN γ may not mainly regulate PD-L1 expression at transcription level, but at translation or protein stability.

Answer: We fully agree with the reviewer that IFN γ induces PD-L1 expression more than transcription level. To verify that, we have treated the SW1990 cells with MG132 to inhibit protein degradation and Actinomycin D to inhibit new mRNA synthesis. And

the upregulation of PD-L1 protein by IFN γ can still be detected (below), suggesting a transcriptional-independent PD-L1 regulation by IFN γ .

9. In Fig.3a, the PD-L1 regulation by IFN γ is not consistent. In MCF-7 and HepG2 cells, the fold change of PD-L1 induced by IFN γ is not affected by siMTHFD2, suggesting that MTHFD2 is not involved in the regulation of PD-L1 by IFN γ . Furthermore, in Fig.3d, the effect of LY294002 on PD-L1 is weak.

Answer: Thanks for this point. In Fig.3a, the PD-L1 mRNA expression obviously decreased after MTHFD2 KD in comparing the siCon/IFN γ and siMTHFD2/IFN γ group, so, we state that the MTHFD2 is involved in IFN γ -induced PD-L1 expression. The same as comments 6, MTHFD2-independent mechanisms like STAT1 signaling are also involved in IFN γ regulating PD-L1. So, that's reasonable that IFN γ can also induce a certain level of PD-L1 expression in MTHFD2 KD cells.

In Fig.3, the effect of AKT inhibitor LY294002 on PD-L1 and MTHFD2 was very convincing in different cell lines in all our repeated experiments. Getting lower basal level usually will get more clear regulation information when performing the immunoblotting experiments, such as another repeat (a little bit dirty but with much lower exposed basal level) in the revised Fig.3c (left only).

10. How is MTHFD2 regulated by AKT/mTOR pathway?

Answer: We apologize for the poor explanation for this point. In the cited report (Science. 2016 Feb 12;351(6274):728-733) in the manuscript, it was indicated that mTOR pathway induces MTHFD2 via ATF4. The abstract of this paper was cited below: ".....mTORC1 had transcriptional effects on multiple enzymes contributing to purine synthesis, with expression of the mitochondrial tetrahydrofolate cycle enzyme methylenetetrahydrofolate dehydrogenase 2 (MTHFD2) being closely associated with mTORC1 signaling in both normal and cancer cells. MTHFD2 expression and purine synthesis were stimulated by activating transcription factor 4 (ATF4), which was activated by mTORC1....."

11. The authors have completely ignored the importance of PD-L1 (B7-H1) in antigen presenting cells, particularly macrophages and dendritic cells.

Answer: We fully agree with the reviewer that these antigen presenting cells (APCs) are important for antigen-specific activation of CD8⁺ T cells *in vivo*. Definitely, the PD-L1 in APCs is very important for the suppression of CD8⁺ T activation. However, the PD-L1 in tumor cells are also involved in T cell inhibition, for example an abstract of a PD-L1 review (Immunity. 2018 Mar 20;48(3):434-452), here: "Expression of programmed death-ligand 1 (PD-L1) is frequently observed in human cancers.....and thus expression of PD-L1 on tumor cells and other cells in the tumor microenvironment is of major clinical relevance." Moreover, the expression of MTHFD2 in dendritic cells is very low. Due to the limitation and the focus of our study (high MTHFD2 in various tumor cells), we are sorry that the important role of PD-L1 in APCs has not been analyzed here.

Reviewer #4 (Remarks to the Author); expert on one-carbon metabolism and immunology:

The study 'Metabolic control of immune evasion via tumour MTHFD2' identifies the folate cycle enzyme MTHFD2 as a factor that can dampen effector T cell cytotoxicity and therefore fosters tumor growth. The authors show that MTHFD2 supports PD1L expression by a mechanism that involves O-GlcNAcylation of cMYC.

This is an interesting paper and most of the experiments are well performed. I especially would like to highlight that the key findings of the current study are supported by various 'rescue experiments' (mutants, treatment with UDP/UTP, inhibitors, reexpression) basically on every level. In addition, findings by the authors are supported by correlations found by the analyses of patient material.

Answer: We greatly appreciate the reviewer's acknowledgement of the significance of our report and the insightful comments that will certainly strengthen our manuscript.

Specific points:

- In figure 3d – 3e, the authors should add IFN γ with Rapamycin and/or LY294002 on the MTHFD2 KD and KO cells to show that the inhibitors do not have an effect of PD-

L1 and that this effect is dependent on the presence of MTHFD2.

Answer: This question is well taken. We have performed the experiment based on the reviewer's suggestion in the revised manuscript (Fig. S3f). Rapamycin obviously inhibited all the IFN- γ -induced MTHFD2, PD-L1 as well as phosphorylated STAT1 (p-STAT1); while PD-L1 was also slightly suppressed by rapamycin in MTHFD2 KD cells, suggesting mTORC1 stimulates PD-L1 only partially through MTHFD2 while STAT1 could be one of mTORC1 downstream events. However, p-STAT1 has not been notably affected by MTHFD2 KD, suggesting STAT1, the canonical downstream of IFN- γ , was neither upstream nor downstream of MTHFD2.

According to our data related to IFN γ (Fig. 3), we emphasize several points about the relationship among IFN γ , AKT/mTOR, MTHFD2 and PD-L1 with the diagram below.

① The basal tumor MTHFD2 upregulating PD-L1 is the key finding in this study (red in the diagram). ② IFN γ upregulating MTHFD2 was an unexpected finding during we analyzing the role of MTHFD2 in IFN γ -induced PD-L1 expression. ③ MTHFD2 KD suppressed both basal and IFN- γ -elevated PD-L1, indicating high basal level of tumor MTHFD2 plus IFN- γ -elevated MTHFD2 both contribute to PD-L1 expression during IFN- γ stimulation. ④ IFN- γ induced notable PD-L1 expression within 4h, but MTHFD2 required 12-24h. This suggests that tumor basal MTHFD2 but not IFN- γ -elevated MTHFD2 is involved in the initial process of IFN- γ -induced PD-L1 expression and the IFN γ -elevated MTHFD2 is just one of the contributors for PD-L1 expression. ⑤ mTORC1 pathway promotes PD-L1 expression highly but not fully dependent on MTHFD2. ⑥ STAT1, the canonical downstream of IFN- γ and known to promotes PD-L1 expression, was neither major upstream nor major downstream of MTHFD2. The MTHFD2 independent ways (purple in the diagram) for IFN γ regulating PD-L1 are also important but out of the focus (MTHFD2) in this study.

- In figure 5a, siControl5 and siControl6 look completely different than the rest of the controls, why is that?

Answer: We thank the reviewer for a careful review. Although we detected a high and significant change between siControl and siMTHFD2 groups for these Uridine related metabolites. Definitely, siControl5 and siControl6 look different compared to the rest of the controls. The cells were prepared and the MTHFD2 levels were confirmed in our hand. But the following procedures for metabolites collection and mass spectrometry (MS) were independently performed by MS experts in other group. After discussing this point with them, we were told that's usual to detect the variation in metabolite samples and the quality of our samples fit the standard before getting on the MS machine. We are very sorry that we are not sure the real reason for that. Therefore, we searched some literatures mainly discussing the sources of variation in metabolomic studies. Below, a attached graph describes some of the possibilities for that.

(Figure.1 Anal. Chem. 2015, 87, 7, 3606–3615)

The original raw data for the metabolomic mass spectrometry has been deposited in the iProX database with password: QgdP. (<https://www.iprox.org/page/DSV021.html?url=1608983741533aWcU>)

- The study would benefit from more physiological data that shows the significance of O-GlcNAcylation. How does expression of Myc(tag)-OGT expression (in tumors) affect T cell killing?...and/or even tumor growth.

Answer: Thanks for this good suggestion. We have performed the experiment based on the reviewer's suggestion in the revised manuscript (Fig. 5g). T cell cytotoxicity assay indicated aggravated cytolysis in MTHFD2 KD cells was significantly reversed by the exogenously expressed OGT, which supports the role of tumor O-GlcNAcylation in affecting T cell killing.

- In the text related to figure 5, the authors conclude that the MTHFD2-enhanced PD-L1 expression is mediated by O-GlcNAcylation. However, to confirm that, they should inhibit O-GlcNAcylation (blocking OGT) and then see if the expression of PD-L1 is affected.

Answer: This question is well taken. We have performed the experiment based on the reviewer's suggestion in the revised manuscript (Fig. 5f). It should be noted that high O-GlcNAcylation promotes cell growth and proliferation, while basal O-GlcNAcylation is essential for cell survival. Fully depleting OGT will cause severe cell death. Here, partially OGT depletion by shRNA obviously inhibited MTHFD2-enhanced PD-L1 expression, which supported the conclusion that the MTHFD2-enhanced PD-L1 expression is mediated by O-GlcNAcylation.

- In figure 7g, they use adenovirus treatment as a potential treatment strategy. I found this a bit stretched and the authors do not really need this data.

Answer: We tried to establish a MTHFD2-targeted strategy for cancer treatment. However, during this process, we found the MTHFD2 is also involved in T cell function. Therefore, we have generated a practical strategy, which only inhibits tumor MTHFD2 but not T cell MTHFD2. We were very excited that this strategy exhibited a promising effect in human tumors under humanized immunity *in vivo*. Although a little pity, we agree with the reviewer's opinion that Fig. 7e-h (Fig. 7e, 7f and 7h tightly related to Fig.

7g) are not tightly related to our general focus that "Metabolic control of immune evasion via tumour MTHFD2" and may be confusing to the reader. Therefore, we have deleted the Fig. 7e-h, and re-organized Fig. 7 in the revised manuscript.

Reviewers' comments:

Reviewer #1 (Remarks to the Author):

The authors have sufficiently addressed my comments.

Reviewer #2 (Remarks to the Author):

The CRISPR screen data have been corrected and their data analysis now looks fine. Although the statistical value of MTHFD2 is quite weak, the selection process of candidate genes and, more importantly, validation of MTHFD2 are reasonably performed and convincing enough to further follow up this gene. I am therefore happy with the screening experiment part.

Reviewer #3 (Remarks to the Author):

The authors have addressed previous comments.

Reviewer #4 (Remarks to the Author):

The authors addressed or discussed all my points.